# Streaming Factor Trajectory Learning for Temporal Tensor Decomposition

**Shikai Fang**
Kahlert School of Computing
The University of Utah
`shikai.fang@utah.edu`

**Xin Yu**
Kahlert School of Computing
The University of Utah
`yuxwind@gmail.com`

**Shibo Li**
Kahlert School of Computing
The University of Utah
`shiboli.cs@gmail.com`

**Zheng Wang**
Kahlert School of Computing
The University of Utah
`u1208847@utah.edu`

**Robert M. Kirby**
Kahlert School of Computing
The University of Utah
`kirby@cs.utah.edu`

**Shandian Zhe**[*]
Kahlert School of Computing
The University of Utah
`zhe@cs.utah.edu`

## Abstract

Practical tensor data is often along with time information. Most existing temporal decomposition approaches estimate a set of fixed factors for the objects in each tensor mode, and hence cannot capture the temporal evolution of the objects' representation. More important, we lack an effective approach to capture such evolution from streaming data, which is common in real-world applications. To address these issues, we propose Streaming Factor Trajectory Learning (SFTL) for temporal tensor decomposition. We use Gaussian processes (GPs) to model the trajectory of factors so as to flexibly estimate their temporal evolution. To address the computational challenges in handling streaming data, we convert the GPs into a state-space prior by constructing an equivalent stochastic differential equation (SDE). We develop an efficient online filtering algorithm to estimate a decoupled running posterior of the involved factor states upon receiving new data. The decoupled estimation enables us to conduct standard Rauch-Tung-Striebel smoothing to compute the full posterior of all the trajectories in parallel, without the need for revisiting any previous data. We have shown the advantage of SFTL in both synthetic tasks and real-world applications. The code is available at `https://github.com/xuangu-fang/Streaming-Factor-Trajectory-Learning`.

## 1  Introduction

Tensor data is common in real-world applications. For example, one can extract a three-mode tensor *(patient, drug, clinic)* from medical service records and a four-mode tensor *(customer, commodity, seller, web-page)* from the database of an online shopping platform. Tensor decomposition is a fundamental tool for tensor data analysis. It introduces a set of factors to represent the objects in each mode, and estimate these factors by reconstructing the observed entry values. These factors can be viewed as the underlying properties of the objects. We can use them to search for interesting structures within the objects (*e.g.,* communities and outliers) or as discriminate features for predictive tasks, such as personalized treatment or recommendation.

Real-world tensor data is often accompanied with time information, namely the timestamps at which the objects of different modes interact to produce the entry values. Underlying the timestamps can be rich, valuable temporal patterns. While many temporal decomposition methods are available,

---

[*]Corresponding author.

37th Conference on Neural Information Processing Systems (NeurIPS 2023).

most of them estimate a set of static factors for each object — they either introduce a discrete time mode [Xiong et al., 2010, Zhe et al., 2016a] or inject the timestamp into the decomposition model [Zhang et al., 2021, Fang et al., 2022, Li et al., 2022]. Hence, these methods cannot learn the temporal variation of the factors. Accordingly, they can miss important evolution of the objects' inner properties, such as health and income. In addition, practical applications produce data streams at a rapid pace [Du et al., 2018]. Due to the resource limit, it is often prohibitively expensive to decompose the entire tensor from scratch whenever we receive new data. Many privacy-protecting applications (*e.g.,* SnapChat) even forbid us to preserve or re-access the previous data. Therefore, not only do we need a more powerful decomposition model that can estimate the factor evolution, we also need an effective method to capture such evolution from fast data streams.

To address these issues, we propose SFTL, a Bayesian streaming factor trajectory learning approach for temporal tensor decomposition. Our method can efficiently handle data streams, with which to estimate a posterior distribution of the factor trajectory to uncover the temporal evolution of the objects' representation. Our method never needs to keep or re-visit previous data. The contribution of our work is summarized as follows.

- First, we use a Gaussian process (GP) prior to sample the factor trajectory of each object as a function of time. As a nonparametric function prior, GPs are flexible enough to capture a variety of complex temporal dynamics. The trajectories are combined through a CANDECOMP/PARAFAC (CP) [Harshman, 1970] or Tucker decomposition [Tucker, 1966] form to sample the tensor entry values at any time point.
- Second, to sidestep the expensive covariance matrix computation in the GP, which further causes challenges in streaming inference, we use spectral analysis to convert the GP into a linear time invariant (LTI) stochastic differential equation (SDE). Then we convert the SDE into an equivalent state-space prior over the factor (trajectory) states at the observed timestamps. As a result, the posterior inference becomes easier and computationally efficient.
- Third, we take advantage of the chain structure of the state-space prior and use the recent conditional expectation propagation framework [Wang and Zhe, 2019] to develop an efficient online filtering algorithm. Whenever a collection of entries at a new timestamp arrives, our method can efficiently estimate a decoupled running posterior of the involved factor states, with a Gaussian product form. The decoupled Gaussian estimate enables us to run standard RTS smoothing [Särkkä, 2013] to compute the full posterior of each factor trajectory independently and in parallel, without revisiting any previous data. Our method at worst has a linear scalability with respect to the number of observed timestamps.

We first evaluated our method in a simulation study. On synthetic datasets, SFTL successfully recovered several nonlinear factor trajectories, and provided reasonable uncertainty estimation. We then tested our method on four real-world temporal tensor datasets for missing value prediction. In both online and final predictive performance, SFTL consistently outperforms the state-of-the-art streaming CP and Tucker decomposition algorithms by a large margin. In most cases, the prediction accuracy of SFTL is even higher than the recent static decomposition methods, which have to pass through the dataset many times. Finally, we investigated the learned factor trajectories from a real-world dataset. The trajectories exhibit interesting temporal evolution.

## 2   Preliminaries

**Tensor Decomposition.** We denote an $M$-mode tensor by $\mathcal{Y} \in \mathbb{R}^{d_1 \times \cdots \times d_M}$, where each mode $m$ has $d_m$ dimensions, corresponding to $d_m$ objects. Each tensor entry is indexed by a tuple $\boldsymbol{\ell} = (\ell_1, \ldots, \ell_M)$, and the value is denoted by $y_{\boldsymbol{\ell}}$. For decomposition, we introduce a set of latent factors $\mathbf{u}_j^m \in \mathbb{R}^{R_m}$ to represent each object $j$ in mode $m$ ($1 \leq m \leq M$). One most popular tensor decomposition model is the CANDECOMP/PARAFAC (CP) decomposition [Harshman, 1970], which sets $R_1 = \ldots = R_M = R$, and uses the following element-wise form, $y_{\boldsymbol{\ell}} \approx \mathbf{1}^\top (\mathbf{u}_{\ell_1}^1 \circ \ldots \circ \mathbf{u}_{\ell_M}^M) = \sum_{r=1}^R \prod_{m=1}^M u_{\ell_m,r}^m$, where $\circ$ is the element-wise product. Another commonly used model is Tucker decomposition [Tucker, 1966], $y_{\boldsymbol{\ell}} \approx \text{vec}(\mathcal{W})^\top (\mathbf{u}_{\ell_1}^1 \otimes \ldots \otimes \mathbf{u}_{\ell_M}^M) = \sum_{r_1=1}^{R_1} \cdots \sum_{r_M=1}^{R_M} \left[ w_{\mathbf{r}} \cdot \prod_{m=1}^M u_{\ell_m,r_m}^m \right]$, where $\mathcal{W} \in \mathbb{R}^{R_1 \times \cdots \times R_M}$ is the tensor-core parameter, $\text{vec}(\cdot)$ is the vectorization, $\otimes$ is the Kronecker product, and $\mathbf{r} = (r_1, \ldots, r_M)$.

**Gaussian Process (GP)s** are nonparametric function priors. For a function $f(\mathbf{x})$, if we place a GP prior, $f \sim \mathcal{GP}(0, \kappa(\mathbf{x}, \mathbf{x}'))$, it means $f(\cdot)$ is sampled as a realization of the GP with covariance function $\kappa$, which is often chosen as a kernel function. The GP prior only models the correlation between the function values, namely, $\text{cov}(f(\mathbf{x}), f(\mathbf{x}')) = \kappa(\mathbf{x}, \mathbf{x}')$, and does not assume any parametric form of the function. Hence, GPs are flexible enough to estimate various complex functions from data, *e.g.,* from multilinear to highly nonlinear. The finite projection of the GP is a Gaussian distribution. That is, given an arbitrary collection of inputs $\{\mathbf{x}_1, \ldots, \mathbf{x}_N\}$, the corresponding function values $\mathbf{f} = [f(\mathbf{x}_1), \ldots, f(\mathbf{x}_N)]^\top$ follow a multi-variate Gaussian prior distribution, $p(\mathbf{f}) = \mathcal{N}(\mathbf{f}|\mathbf{0}, \mathbf{K})$ where $\mathbf{K}$ is the covariance matrix and each $[\mathbf{K}]_{mn} = \kappa(\mathbf{x}_m, \mathbf{x}_n)$.

## 3 Bayesian Temporal Tensor Decomposition with Factor Trajectories

In real-world applications, tensor data is often associated with time information, namely, the timestamps at which the objects of different modes interact to generate the entry values. To capture the potential evolution of the objects' inner properties, we propose a Bayesian temporal tensor decomposition model that can estimate a trajectory of the factor representation. Specifically, for each object $j$ in mode $m$, we model the factors as a function of time, $\mathbf{u}_j^m : [0, \infty] \to \mathbb{R}^R$. To flexibly capture a variety of temporal evolution, we assign a GP prior over each element of $\mathbf{u}_j^m(t) = [u_{j,1}^m(t), \ldots, u_{j,R}^m(t)]^\top$, *i.e.,* $u_{j,r}^m(t) \sim \mathcal{GP}(0, \kappa(t, t'))$ $(1 \leq r \leq R)$. Given the factor trajectories, we then use the CP or Tucker form to sample the entry values at different time points. For the CP form, we have

$$p(y_{\boldsymbol{\ell}}(t)|\mathcal{U}(t)) = \mathcal{N}(y_{\boldsymbol{\ell}}(t)|\mathbf{1}^\top(\mathbf{u}_{\ell_1}^1(t) \circ \ldots \circ \mathbf{u}_{\ell_M}^M(t)), \tau^{-1}), \tag{1}$$

where $\mathcal{U}(t) = \{\mathbf{u}_j^m(t)\}$ includes all the factor trajectories, and $\tau$ is the inverse noise variance, for which we assign a Gamma prior, $p(\tau) = \text{Gam}(\tau|\alpha_0, \alpha_1)$. For the Tucker form, we have $p(y_{\boldsymbol{\ell}}(t)|\mathcal{U}(t), \mathcal{W}) = \mathcal{N}(y_{\boldsymbol{\ell}}(t)|\text{vec}(\mathcal{W})^\top(\mathbf{u}_{\ell_1}^1(t) \otimes \ldots \otimes \mathbf{u}_{\ell_M}^M(t)), \tau^{-1})$ where we place a standard normal prior over the tensor-core, $p(\text{vec}(\mathcal{W})) = \mathcal{N}(\text{vec}(\mathcal{W})|\mathbf{0}, \mathbf{I})$. In this work, we focus on continuous observations. It is straightforward to extend our method for other types of observations.

Suppose we have a collection of observed entry values and timestamps, $\mathcal{D} = \{(\boldsymbol{\ell}_1, y_1, t_1), \ldots, (\boldsymbol{\ell}_N, y_N, t_N)\}$ where $t_1 \leq \cdots \leq t_N$. We denote the sequence of timestamps when a particular object $j$ of mode $m$ participated in the observed entries by $s_{j,1}^m < \ldots < s_{j,c_j^m}^m$, where $c_j^m$ is the participation count of the object. Note that it is a sub-sequence of $\{t_n\}$. From the GP prior, the values of each $u_{j,r}^m(t)$ at these timestamps follow a multi-variate Gaussian distribution, $p(\mathbf{u}_{j,r}^m) = \mathcal{N}(\mathbf{u}_{j,r}^m|\mathbf{0}, \mathbf{K}_j^m)$ where $\mathbf{u}_{j,r}^m = [u_{j,r}^m(s_{j,1}^m), \ldots, u_{j,r}^m(s_{j,c_j^m}^m)]^{\top 2}$ and $\mathbf{K}_j^m$ is the covariance/kernel matrix computed at these timestamps. The joint probability with the CP form is

$$p(\{\mathbf{u}_{j,r}^m\}, \tau, \mathbf{y}) = \prod_{m=1}^M \prod_{j=1}^{d_m} \prod_{r=1}^R \mathcal{N}(\mathbf{u}_{j,r}^m|\mathbf{0}, \mathbf{K}_j^m) \cdot \text{Gam}(\tau|\alpha_0, \alpha_1)$$

$$\cdot \prod_{n=1}^N \mathcal{N}(y_n|\mathbf{1}^\top(\mathbf{u}_{\ell_{n1}}^1(t_n) \circ \ldots \circ \mathbf{u}_{\ell_{nM}}^M(t_n)), \tau^{-1}). \tag{2}$$

The joint probability with the Tucker form is the same except that we use the Tucker likelihood instead and multiply with the prior of tensor-core $p(\mathcal{W})$.

While this formulation is straightforward, it can introduce computational challenges. There are many multi-variate Gaussian distributions in the joint distribution (2), *i.e.,* $\{\mathcal{N}(\mathbf{u}_{j,r}^m|\mathbf{0}, \mathbf{K}_j^m)\}$. The time and space complexity to compute each $\mathcal{N}(\mathbf{u}_{j,r}^m|\mathbf{0}, \mathbf{K}_j^m)$ is $\mathcal{O}\left((c_j^m)^3\right)$ and $\mathcal{O}\left((c_j^m)^2\right)$, respectively. With the increase of $N$, the appearance count $c_j^m$ for many objects can grow as well, making the computation cost very expensive or even infeasible. The issue is particularly severe when we handle streaming data — the number of timestamps grows rapidly when new data keeps coming in, so does the size of each covariance matrix.

### 3.1 Equivalent Modeling with State-Space Priors

To sidestep expensive covariance matrix computation and ease the inference with streaming data, we follow [Hartikainen and Särkkä, 2010] to convert the GP prior into an SDE via spectral analysis.

---

[2]For convenience, we abuse the notation a little bit: we denote by $\mathbf{u}_{j,r}^m(\cdot)$ trajectory function and by $\mathbf{u}_{j,r}^m$ the values of the trajectory function at the observed timestamps.

We use a Matérn kernel $\kappa_\nu(t, t') = a \frac{\left(\frac{\sqrt{2\nu}}{\rho}\Delta\right)^\nu}{\Gamma(\nu)2^{\nu-1}} K_\nu\left(\frac{\sqrt{2\nu}}{\rho}\Delta\right)$ where $\Gamma(\cdot)$ is the Gamma function, $\Delta = |t - t'|$, $a > 0$, $\rho > 0$, $K_\nu$ is the modified Bessel function of the second kind, and $\nu = p + \frac{1}{2}$ ($p \in \{0, 1, 2, \ldots\}$) as the GP covariance. Via the analysis of the power spectrum of $\kappa_\nu$, we can show that if $f(t) \sim \mathcal{GP}(0, \kappa_\nu(t, t'))$, it can be characterized by a linear time-invariant (LTI) SDE, with state $\mathbf{z} = (f, f^{(1)}, \ldots, f^{(p)})^\top$ where $f^{(k)} \triangleq \mathrm{d}^k f / \mathrm{d}t^k$,

$$\frac{\mathrm{d}\mathbf{z}}{\mathrm{d}t} = \mathbf{A}\mathbf{z} + \boldsymbol{\eta} \cdot \beta(t), \tag{3}$$

where $\beta(t)$ is a white noise process with diffusion $\sigma^2$,

$$\mathbf{A} = \begin{pmatrix} 0 & 1 & & \\ & \ddots & \ddots & \\ & & 0 & 1 \\ -c_0 & \ldots & -c_{p-1} & -c_p \end{pmatrix}, \quad \boldsymbol{\eta} = \begin{pmatrix} 0 \\ \vdots \\ 0 \\ 1 \end{pmatrix}.$$

Both $\sigma^2$ and $\mathbf{A}$ are obtained from the parameters in $\kappa_\nu$. Due to the space limit, we leave the detailed derivation in Appendix (Section A). The LTI-SDE is particularly useful in that its finite set of states follow a Gauss-Markov chain, *i.e.,* the state-space prior. Given arbitrary $t_1 < \ldots < t_L$, we have

$$p(\mathbf{z}(t_1), \ldots, \mathbf{z}(t_L)) = p(\mathbf{z}(t_1)) \prod\nolimits_{k=1}^{L-1} p(\mathbf{z}(t_{k+1})|\mathbf{z}(t_k)),$$

where $p(\mathbf{z}(t_1)) = \mathcal{N}(\mathbf{z}(t_1)|\mathbf{0}, \mathbf{P}_\infty)$, $p(\mathbf{z}(t_{k+1})|\mathbf{z}(t_k)) = \mathcal{N}(\mathbf{z}(t_{k+1})|\mathbf{F}_k\mathbf{z}(t_k), \mathbf{Q}_k)$, $\mathbf{P}_\infty$ is the stationary covariance matrix computed by solving the matrix Riccati equation [Lancaster and Rodman, 1995], $\mathbf{F}_n = \exp(\Delta_k \cdot \mathbf{A})$ where $\Delta_k = t_{k+1} - t_k$, and $\mathbf{Q}_k = \mathbf{P}_\infty - \mathbf{A}_k\mathbf{P}_\infty\mathbf{A}_k^\top$. Therefore, we do not need the full covariance matrix as in the standard GP prior, and the computation is much more efficient. The chain structure is also convenient to handle streaming data as we will explain later.

We therefore convert the GP prior over each factor trajectory $u_{j,r}^m(t)$ into an LTI-SDE. We denote the corresponding state by $\mathbf{z}_{j,r}^m(t)$. For example, if we choose $p = 1$, then $\mathbf{z}_{j,r}^m(t) = [u_{j,r}^m(t); \mathrm{d}u_{j,r}^m(t)/\mathrm{d}t]$. For each object $j$ in mode $m$, we concatenate all its trajectory states into one, $\mathbf{z}_j^m(t) = [\mathbf{z}_{j,1}^m(t); \ldots; \mathbf{z}_{j,R}^m(t)]$. Then on all of its timestamps $s_{j,1}^m < \ldots < s_{j,c_j^m}^m$, we obtain a state-space prior

$$p(\mathbf{h}_{j,1}^m) = \mathcal{N}(\mathbf{h}_{j,1}^m|\mathbf{0}, \overline{\mathbf{P}}_\infty), \quad p(\mathbf{h}_{j,k+1}^m|\mathbf{h}_{j,k}^m) = \mathcal{N}(\mathbf{h}_{j,k+1}^m|\overline{\mathbf{F}}_{j,k}^m\mathbf{h}_{j,k}^m, \overline{\mathbf{Q}}_{j,k}^m), \tag{4}$$

where $\mathbf{h}_{j,k}^m \triangleq \mathbf{z}_j^m(s_{j,k}^m)$, $\overline{\mathbf{P}}_\infty = \mathrm{diag}(\mathbf{P}_\infty, \ldots, \mathbf{P}_\infty)$, $\overline{\mathbf{F}}_{j,k}^m = \mathrm{diag}\left(\mathbf{F}_{j,k}^m, \ldots, \mathbf{F}_{j,k}^m\right)$, $\mathbf{F}_{j,k}^m = e^{(s_{j,k+1}^m - s_{j,k}^m)\mathbf{A}}$, $\overline{\mathbf{Q}}_{j,k}^m = \mathrm{diag}\left(\mathbf{Q}_{j,k}^m, \ldots, \mathbf{Q}_{j,k}^m\right)$, and $\mathbf{Q}_{j,k}^m = \mathbf{P}_\infty - \mathbf{F}_{j,k}^m\mathbf{P}_\infty\left(\mathbf{F}_{j,k}^m\right)^\top$.

The joint probability of our model with the CP form now becomes

$$p(\{\mathbf{h}_{j,k}^m\}, \tau, \mathbf{y}) = p(\tau) \prod\nolimits_{m=1}^M \prod\nolimits_{j=1}^{d_m} p(\mathbf{h}_{j,1}^m) \prod\nolimits_{k=1}^{c_j^m-1} p(\mathbf{h}_{j,k+1}^m|\mathbf{h}_{j,k}^m)$$

$$\cdot \prod\nolimits_{n=1}^N \mathcal{N}(y_n|\mathbf{1}^\top(\mathbf{u}_{\ell_{n1}}^1(t_n) \circ \ldots \circ \mathbf{u}_{\ell_{nM}}^M(t_n)), \tau^{-1}). \tag{5}$$

Note that in the likelihood, each $\mathbf{u}_{\ell_{nm}}^m(t_n)(1 \leq j \leq M)$ is contained in a corresponding state vector $\mathbf{h}_{\ell_{nm},k}^m$ such that $s_{\ell_{nm},k}^m = t_n$ (by definition, we then have $\mathbf{h}_{\ell_{nm},k}^m = \mathbf{z}_{\ell_{nm}}^m(t_n)$). The joint probability with the Tucker form is similar, which we omit to save the space.

## 4   Trajectory Inference from Streaming Data

In this section, we develop an efficient, scalable algorithm for factor trajectory estimation from streaming data. In general, we assume that we receive a sequence of (small) batches of observed tensor entries, $\{\mathcal{B}_1, \mathcal{B}_2, \ldots\}$, generated at different timestamps, $\{t_1, t_2, \ldots\}$. Each batch $\mathcal{B}_n$ is generated at timestamp $t_n$ and $t_n < t_{n+1}$. Denote by $\mathcal{D}_{t_n}$ all the data up to timestamp $t_n$, *i.e.,* $\mathcal{D}_{t_n} = \mathcal{B}_1 \cup \ldots \cup \mathcal{B}_n$. Upon receiving $\mathcal{B}_{n+1}$, we intend to update our model without revisiting $\mathcal{D}_{t_n}$

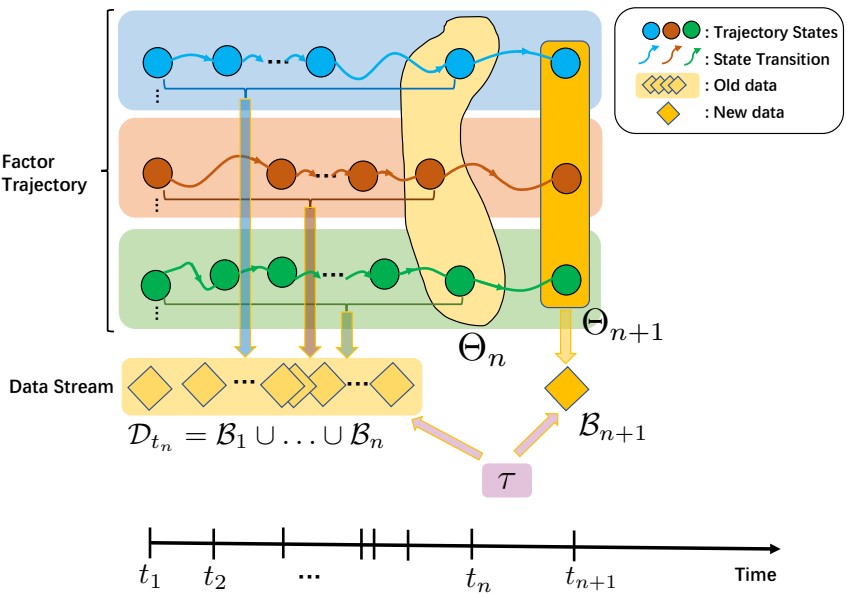

Figure 1: A graphical representation of our factor trajectory learning, from which we can see $\{\Theta_{n+1}, \mathcal{B}_{n+1}\}$ are independent to $\mathcal{D}_{t_n}$ conditioned on $\Theta_n$ and the noise inverse variance $\tau$, namely, $\Theta_{n+1}, \mathcal{B}_{n+1} \perp \mathcal{D}_{t_n} | \Theta_n, \tau$.

to provide the trajectory posterior estimate, $\{p(\mathbf{u}_j^m(t)|\mathcal{D}_{t_{n+1}})|\forall t \geq 0, 1 \leq m \leq M, 1 \leq j \leq d_m\}$, where $\mathcal{D}_{t_{n+1}} = \mathcal{D}_{t_n} \cup \mathcal{B}_{n+1}$.

To this end, we first observe that the standard state-space model with a Gaussian likelihood has already provided a highly efficient streaming inference framework. Denote by $\mathbf{x}_n$ and $\mathbf{y}_n$ the state and observation at each step $n$, respectively. To handle streaming observations $\{\mathbf{y}_1, \mathbf{y}_2, \ldots\}$, we only need to compute and track the running posterior $p(\mathbf{x}_n|\mathbf{y}_{1:n})$ upon receiving each $\mathbf{y}_n$, where $\mathbf{y}_{1:n}$ denotes the total data up to step $n$. This is called Kalman filtering [Särkkä, 2013], which only depends on the running posterior at step $n-1$, *i.e.,* $p(\mathbf{x}_{n-1}|\mathbf{y}_{1:n-1})$, and is highly efficient. After all the data is processed (suppose it stops at step $N$), we can use Rauch-Tung-Striebel (RTS) smoother [Särkkä, 2013] to efficiently compute the full posterior of each state, $p(\mathbf{x}_n|\mathbf{y}_{1:N})$, from backward, which does not need to re-access any previous observations (see Section B in Appendix).

However, one cannot apply the above framework outright to our model, since the tensor decomposition likelihood of each observed entry couples the states of multiple factor trajectories, see (1) — which correspond to the participated objects at different modes. That means, the factor-state chains of different objects are dynamically intertwined through the received data. The multiplicative form of these states in the likelihood render the running posterior of each trajectory intractable to compute, not to mention running RTS smoother. To address this challenge, we take advantage of the chain structure and use the recent conditional Expectation propagation (CEP) framework [Wang and Zhe, 2019] to develop an efficient online filtering algorithm, which approximates the running posterior of the involved factor states as a product of Gaussian. Thereby, we can decouple the involved factor state chains, and conduct standard RTS smoothing for each chain independently.

Specifically, denote the sequence of timestamps when each object $j$ of mode $m$ has showed up in the data stream up to $t_n$, by $s_{j,1}^m < s_{j,2}^m < \ldots < s_{j,c_{j,n}^m}^m$ where $c_{j,n}^m$ is the object's appearance count up to $t_n$. Denote by $\mathcal{I}_n^m$ the indexes of all the objects of mode $m$ appearing in $\mathcal{B}_n$. Hence, for every object $j \in \mathcal{I}_n^m$, we have $s_{j,c_{j,n}^m}^m = t_n$, and $\mathbf{h}_{j,c_{j,n}^m}^m \triangleq \mathbf{z}_j^m(t_n)$ is the factor state of the object at $t_n$. Upon receiving each $\mathcal{B}_n$, we intend to approximate the running posterior of all the involved factor states and noise inverse variance $\tau$ with the following decoupled form,

$$p(\tau, \{\mathbf{h}_{j,c_{j,n}^m}^m | j \in \mathcal{I}_n^m\}_{1 \leq m \leq M} | \mathcal{D}_{t_n}) \approx q(\tau | \mathcal{D}_n) \prod_{m=1}^M \prod_{j \in \mathcal{I}_n^m} q(\mathbf{h}_{j,c_{j,n}^m}^m | \mathcal{D}_{t_n}), \qquad (6)$$

where $q(\tau | \mathcal{D}_{t_n}) = \text{Gam}(\tau | a_n, b_n)$, and $q(\mathbf{h}_{j,c_{j,n}^m}^m | \mathcal{D}_{t_n}) = \mathcal{N}(\mathbf{h}_{j,c_{j,n}^m}^m | \widehat{\boldsymbol{\mu}}_{j,c_{j,n}^m}^m, \widehat{\mathbf{V}}_{j,c_{j,n}^m}^m)$. To this end, let us consider given the approximation at $t_n$, how to obtain the new approximation at $t_{n+1}$ (*i.e.,* upon

receiving $\mathcal{B}_{n+1}$) in the same form of (6). To simplify the notation, let us define the preceding states of the involved factors by $\Theta_n = \{\mathbf{h}^m_{j,c^m_{j,n}} | j \in \mathcal{I}^m_{n+1}\}_m$, and the current states by $\Theta_{n+1} = \{\mathbf{h}^m_{j,c^m_{j,n+1}} | j \in \mathcal{I}^m_{n+1}\}_m$. First, due to the chain structure of the prior over each $\{\mathbf{h}^m_{j,c^m_{j,k}} | k = 0, 1, 2, \ldots\}$, we can see that conditioned on $\{\Theta_n, \tau\}$, the current states $\Theta_{n+1}$ and the new observations $\mathcal{B}_{n+1}$ are independent of $\mathcal{D}_{t_n}$. This is because in the graphical model representation, $\{\Theta_n, \tau\}$ have blocked all the paths from the old observations $\mathcal{D}_{t_n}$ to the new state and observations [Bishop, 2007]; see Fig. 1 for an illustration. Then, we can derive that

$$p(\Theta_{n+1}, \Theta_n, \tau | \mathcal{D}_{t_{n+1}}) \propto p(\Theta_{n+1}, \Theta_n, \tau, \mathcal{B}_{n+1} | \mathcal{D}_{t_n}) = p(\Theta_n, \tau | \mathcal{D}_{t_n}) p(\Theta_{n+1}, \mathcal{B}_{n+1} | \Theta_n, \tau, \mathcal{D}_{t_n})$$

$$= p(\Theta_n, \tau | \mathcal{D}_{t_n}) p(\Theta_{n+1} | \Theta_n) p(\mathcal{B}_{n+1} | \Theta_{n+1}, \tau), \tag{7}$$

where $p(\Theta_n, \tau | \mathcal{D}_{t_n})$ is the running posterior at $t_n$,

$$p(\Theta_{n+1} | \Theta_n) = \prod_{m=1}^M \prod_{j \in \mathcal{I}^m_{n+1}} p(\mathbf{h}^m_{j,c^m_{j,n+1}} | \mathbf{h}^m_{j,c^m_{j,n}}),$$

each $p(\mathbf{h}^m_{j,c^m_{j,n+1}} | \mathbf{h}^m_{j,c^m_{j,n}})$ is a conditional Gaussian distribution defined in (4), and $p(\mathcal{B}_{n+1} | \Theta_{n+1}, \tau) = \prod_{(\boldsymbol{\ell}, y) \in \mathcal{B}_{n+1}} \mathcal{N}(y | \mathbf{1}^\top (\mathbf{u}^1_{\ell_1}(t_{n+1}) \circ \cdots \circ \mathbf{u}^M_{\ell_M}(t_{n+1})), \tau^{-1})$. Since $p(\Theta_n, \tau | \mathcal{D}_{t_n})$ takes the form of (6), we can analytically marginalize out each $\mathbf{h}^m_{j,c^m_{j,n}} \in \Theta_n$, and obtain

$$p(\Theta_{n+1}, \tau | \mathcal{D}_{t_{n+1}}) \propto \text{Gam}(\tau | a_n, b_n) \prod_{m=1}^M \prod_{j \in \mathcal{I}^m_{n+1}} \mathcal{N}(\mathbf{h}^m_{j,c^m_{j,n+1}} | \widehat{\boldsymbol{\mu}}^m_{j,c^m_{j,n+1}}, \widehat{\mathbf{V}}^m_{j,c^m_{j,n+1}}) \tag{8}$$

$$\cdot \prod_{(\boldsymbol{\ell}, y) \in \mathcal{B}_{n+1}} \mathcal{N}(y | \mathbf{1}^\top (\mathbf{u}^1_{\ell_1}(t_{n+1}) \circ \cdots \circ \mathbf{u}^M_{\ell_M}(t_{n+1})), \tau^{-1}).$$

If we view the R.H.S of (8) as a joint distribution with $\mathcal{B}_{n+1}$, then our task amounts to estimating the posterior distribution, *i.e.,* the L.H.S of (8). The product in the CP likelihood (and also Tucker likelihood) renders exact posterior computation infeasible, and we henceforth approximate

$$\mathcal{N}(y | \mathbf{1}^\top (\mathbf{u}^1_{\ell_1}(t_{n+1}) \circ \cdots \circ \mathbf{u}^M_{\ell_M}(t_{n+1}))) \mathrel{\widetilde{\propto}} \prod_{m=1}^M \mathcal{N}(\mathbf{u}^m_{\ell_m}(t_{n+1}) | \boldsymbol{\gamma}^m_{\ell_m}, \boldsymbol{\Sigma}^m_{\ell_m}) \text{Gam}(\tau | \alpha_{\boldsymbol{\ell}}, \omega_{\boldsymbol{\ell}}) \tag{9}$$

where $\mathrel{\widetilde{\propto}}$ means approximately proportional to. To optimize these approximation terms, we use the recent conditional Expectation propagation (CEP) framework [Wang and Zhe, 2019] to develop an efficient inference algorithm. It uses conditional moment matching to update each approximation in parallel and conducts fixed point iterations, and hence can converge fast. We leave the details in the Appendix (Section C). Once it is done, we substitute the approximation (9) into (8). Then the R.H.S of (8) becomes a product of Gaussian and Gamma terms over each state and $\tau$. We can then immediately obtain a closed-form estimation in the form as (6). At the beginning, when estimating $p(\Theta_1, \tau | \mathcal{D}_{t_1})$, since the preceding states $\Theta_0 = \emptyset$, we have $a_n = \alpha_0$, $b_n = \alpha_1$, $\widehat{\boldsymbol{\mu}}^m_{j,c^m_{j,n+1}} = \mathbf{0}$, and $\widehat{\mathbf{V}}^m_{j,c^m_{j,n+1}} = \overline{\mathbf{P}}_\infty$ in (8), which is the prior of each $\mathbf{h}^m_{j,c^m_{j,1}}$ and $\tau$ (see (4)).

In this way, we can continuously filter the incoming batches $\{\mathcal{B}_1, \mathcal{B}_2, \ldots\}$. As a result, for the factor state chain of every object $j$ in every mode $m$, along with each timestamp $s^m_{j,k}$, we can online estimate and track a running posterior approximation $\{q(\mathbf{h}^m_{j,k} | \mathcal{D}_{t_{s^m_{j,k}}}) | k = 1, 2, \ldots\}$, which is a Gaussian distribution. Hence, we can run the standard RTS smoother, to compute the full posterior of every factor state, with which we can compute the posterior of the trajectory at any time point $t$ [Bishop, 2006]. Our method is summarized in Algorithm 1.

**Algorithm Complexity.** The time complexity of our algorithm processing a batch $\mathcal{B}_n$ is $\mathcal{O}(|\mathcal{B}_n| R^3)$ where $|\cdot|$ is the size. The time complexity of RTS smoother for a particular object $j$ in mode $m$ is $\mathcal{O}(R^3 c^m_{j,N})$, where $N$ is the total number of timestamps. The space complexity of our algorithm is $\mathcal{O}(\sum_{m=1}^M \sum_{j=1}^{d_m} c^m_{j,N} R^2)$, which is to track the running posterior of the factor state at each appearing timestamp for every object. Since $c^m_{j,N} \leq N$, the complexity of our algorithm is at worst linear in $N$.

## 5   Related Work

Many tensor decomposition methods have been developed, such as [Yang and Dunson, 2013, Rai et al., 2014, Zhe et al., 2015, 2016b,a, Tillinghast et al., 2020, Pan et al., 2020b, Fang et al., 2021a,b,

---

**Algorithm 1** Streaming Factor Trajectory Learning (SFTL)

---

1: **Input:** kernel hyper-parameters $a$, $\rho$, $\nu = p + \frac{1}{2}$ ($p \in \{0, 1, 2, \ldots\}$)
2: $n \leftarrow 0$
3: **while** Receiving a new batch of entreis $\mathcal{B}_{n+1}$ **do**
4:    **if** $n = 0$ **then**
5:       Set $a_n = a_0$, $b_n = b_0$, $\widehat{\boldsymbol{\mu}}^m_{j,c^m_{j,n+1}} = \mathbf{0}$, and $\widehat{\mathbf{V}}^m_{j,c^m_{j,n+1}} = \overline{\mathbf{P}}_\infty$ in (8).
6:       Goto 9.
7:    **end if**
8:    Retrieve the involved preceding factor states $\Theta_n = \{\mathbf{h}^m_{j,c^m_{j,n}} | j \in \mathcal{I}^m_{n+1}\}_m$ and their running posterior, $p(\Theta_n, \tau | \mathcal{D}_{t_n}) \approx \text{Gam}(\tau | a_n, b_n) \prod_{m=1}^M \prod_{j \in \mathcal{I}^m_{n+1}} \mathcal{N}(\mathbf{h}^m_{j,c^m_{j,n}} | \widehat{\boldsymbol{\mu}}^m_{j,c^m_{j,n}}, \widehat{\mathbf{V}}^m_{j,c^m_{j,n}})$.
9:    According to (8) and (9), use conditional Expectation Propagation to calculate the running posterior of the current factor states, $p(\Theta_{n+1}, \tau | \mathcal{D}_{t_{n+1}}) \approx \text{Gam}(\tau | a_{n+1}, b_{n+1}) \prod_{m=1}^M \prod_{j \in \mathcal{I}^m_{n+1}} \mathcal{N}(\mathbf{h}^m_{j,c^m_{j,n+1}} | \widehat{\boldsymbol{\mu}}^m_{j,c^m_{j,n+1}}, \widehat{\mathbf{V}}^m_{j,c^m_{j,n+1}})$.
10:    **if** Needed **then**
11:       Run RTS smoothing on any factor state chain $\{\mathbf{h}^m_{j,k} | k = 1, 2, \ldots\}$ of interest.
12:    **end if**
13:    $n \leftarrow n + 1$
14: **end while**
15: Run RTS smoothing for every factor state chain $\{\mathbf{h}^m_{j,k} | k = 1, 2, \ldots\}$.
16: **Return:** $\{q(\mathbf{h}^m_{j,k} | \mathcal{D}) | k = 1, 2, \ldots\}_{1 \le m \le M, 1 \le j \le d_m}$, $q(\tau | \mathcal{D})$, where $\mathcal{D}$ is all the data received.

---

Tillinghast and Zhe, 2021, Tillinghast et al., 2022, Fang et al., 2022, Zhe and Du, 2018, Pan et al., 2020a, Wang et al., 2020, Pan et al., 2021, Wang et al., 2022]. For temporal decomposition, most existing methods augment the tensor with a discrete time mode to estimate additional factors for time steps, *e.g.,* [Xiong et al., 2010, Rogers et al., 2013, Song et al., 2017, Du et al., 2018, Ahn et al., 2021]. The most recent works have conducted continuous-time decomposition. Zhang et al. [2021] used polynomial splines to model a time function as the CP coefficients. Li et al. [2022] used neuralODE [Chen et al., 2018] to model the entry value as a function of latent factors and time point. Fang et al. [2022] performed continuous-time Tucker decomposition, and modeled the tensor-core as a time function. To our knowledge, [Wang et al., 2022] is the first work to estimate factor trajectories. It places a GP prior in the frequency domain, and samples the factor trajectories via inverse Fourier transform. It then uses another GP to sample the entry values. While successful, this method cannot handle streaming data, and the black-box GP decomposition lacks interpretability.

Current Bayesian streaming tensor decomposition methods include [Du et al., 2018, Fang et al., 2021a, Pan et al., 2020b, Fang et al., 2021b], which are based on streaming variational Bayes [Broderick et al., 2013] or assumed density filtering (ADF) [Boyen and Koller, 1998]. ADF can be viewed as an instance of Expectation Propagation (EP) [Minka, 2001] for streaming data. EP approximates complex terms in the probability distribution with exponential-family members, and uses moment matching to iteratively update the approximations, which essentially is a fixed point iteration. To address the challenge of intractable moment matching, Wang and Zhe [2019] proposed conditional EP (CEP), which uses conditional moment matching and Taylor expansion to compute the moments for factorized approximations. The theoretical guarantees and error bound analysis for EP and ADF have been studied for a long time, such as [Boyen and Koller, 1998, Dehaene and Barthelmé, 2015, 2018]. The most recent work [Fang et al., 2022] also uses SDEs to represent GPs and CEP framework for inference, but their GP prior is placed on the tensor-core, not for learning factor trajectories, and their method is only for static decomposition, and cannot handle streaming data.

## 6 Experiment

### 6.1 Simulation Study

We first conducted a simulation study, for which we simulated a two-mode tensor, with two nodes per mode. Each node is represented by a time-varying factor: $u^1_1(t) = -\sin^3(2\pi t)$, $u^1_2(t) = \left(1 - \sin^3(\frac{1}{2}\pi t)\right) \sin^3(3\pi t)$, $u^2_1(t) = \sin(2\pi t)$, and $u^2_2(t) = -\cos^3(3\pi t) \sin(3\pi t) \sin(2\pi t)$. Given

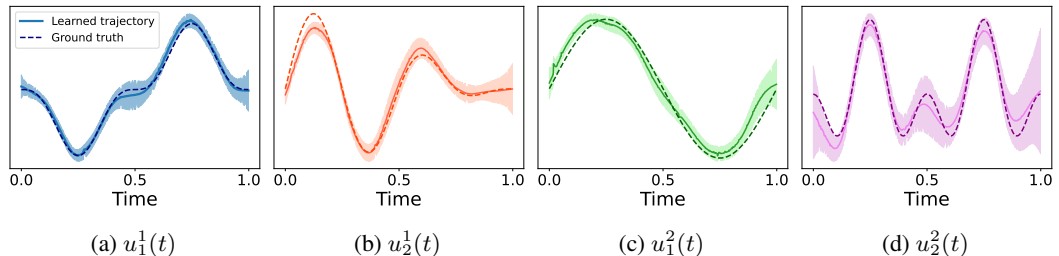

(a) $u_1^1(t)$       (b) $u_2^1(t)$       (c) $u_1^2(t)$       (d) $u_2^2(t)$

Figure 2: The learned factor trajectories from the synthetic data. The shaded region indicates the posterior standard deviation.

| | RMSE | *FitRecord* | *ServerRoom* | *BeijingAir-2* | *BeijingAir-3* |
|---|---|---|---|---|---|
| | PTucker | $0.656 \pm 0.147$ | $0.458 \pm 0.039$ | $0.401 \pm 0.01$ | $0.535 \pm 0.062$ |
| | Tucker-ALS | $0.846 \pm 0.005$ | $0.985 \pm 0.014$ | $0.559 \pm 0.021$ | $0.838 \pm 0.026$ |
| | CP-ALS | $0.882 \pm 0.017$ | $0.994 \pm 0.015$ | $0.801 \pm 0.082$ | $0.875 \pm 0.028$ |
| Static | CT-CP | $0.664 \pm 0.007$ | $0.384 \pm 0.009$ | $0.64 \pm 0.007$ | $0.815 \pm 0.018$ |
| | CT-GP | $0.604 \pm 0.004$ | $0.223 \pm 0.035$ | $0.759 \pm 0.02$ | $0.892 \pm 0.026$ |
| | BCTT | $0.518 \pm 0.007$ | $0.185 \pm 0.013$ | $0.396 \pm 0.022$ | $0.801 \pm 0.02$ |
| | NONFAT | $0.503 \pm 0.002$ | $\mathbf{0.117 \pm 0.006}$ | $0.395 \pm 0.007$ | $0.882 \pm 0.014$ |
| | THIS-ODE | $0.526 \pm 0.004$ | $0.132 \pm 0.003$ | $0.54 \pm 0.014$ | $0.877 \pm 0.026$ |
| | POST | $0.696 \pm 0.019$ | $0.64 \pm 0.028$ | $0.516 \pm 0.028$ | $0.658 \pm 0.103$ |
| | ADF-CP | $0.648 \pm 0.008$ | $0.654 \pm 0.008$ | $0.548 \pm 0.015$ | $0.551 \pm 0.043$ |
| Stream | BASS-Tucker | $0.976 \pm 0.024$ | $1.000 \pm 0.016$ | $1.049 \pm 0.037$ | $0.991 \pm 0.039$ |
| | SFTL-CP | $\mathbf{0.424 \pm 0.014}$ | $0.161 \pm 0.014$ | $\mathbf{0.248 \pm 0.012}$ | $0.473 \pm 0.013$ |
| | SFTL-Tucker | $0.430 \pm 0.010$ | $0.331 \pm 0.056$ | $0.303 \pm 0.041$ | $\mathbf{0.439 \pm 0.019}$ |
| | MAE | | | | |
| | PTucker | $0.369 \pm 0.009$ | $0.259 \pm 0.008$ | $0.26 \pm 0.006$ | $0.263 \pm 0.02$ |
| | Tucker-ALS | $0.615 \pm 0.006$ | $0.739 \pm 0.008$ | $0.388 \pm 0.008$ | $0.631 \pm 0.017$ |
| | CP-ALS | $0.642 \pm 0.012$ | $0.746 \pm 0.009$ | $0.586 \pm 0.056$ | $0.655 \pm 0.018$ |
| Static | CT-CP | $0.46 \pm 0.004$ | $0.269 \pm 0.003$ | $0.489 \pm 0.006$ | $0.626 \pm 0.01$ |
| | CT-GP | $0.414 \pm 0.001$ | $0.165 \pm 0.034$ | $0.55 \pm 0.012$ | $0.626 \pm 0.011$ |
| | BCTT | $0.355 \pm 0.005$ | $0.141 \pm 0.011$ | $0.254 \pm 0.007$ | $0.578 \pm 0.009$ |
| | NONFAT | $0.341 \pm 0.001$ | $\mathbf{0.071 \pm 0.004}$ | $0.256 \pm 0.004$ | $0.626 \pm 0.007$ |
| | THIS-ODE | $0.363 \pm 0.004$ | $0.083 \pm 0.002$ | $0.345 \pm 0.004$ | $0.605 \pm 0.013$ |
| | POST | $0.478 \pm 0.014$ | $0.476 \pm 0.023$ | $0.352 \pm 0.022$ | $0.486 \pm 0.095$ |
| | ADF-CP | $0.449 \pm 0.006$ | $0.496 \pm 0.007$ | $0.385 \pm 0.012$ | $0.409 \pm 0.029$ |
| Stream | BASS | $0.772 \pm 0.031$ | $0.749 \pm 0.01$ | $0.934 \pm 0.037$ | $0.731 \pm 0.02$ |
| | SFTL-CP | $\mathbf{0.242 \pm 0.006}$ | $0.108 \pm 0.008$ | $\mathbf{0.15 \pm 0.003}$ | $0.318 \pm 0.008$ |
| | SFTL-Tucker | $0.246 \pm 0.001$ | $0.216 \pm 0.034$ | $0.185 \pm 0.029$ | $\mathbf{0.278 \pm 0.011}$ |

Table 1: Final prediction error with $R = 5$. The results were averaged from five runs.

these factors, an entry value at time $t$ is generated via $y_{(i,j)}(t) \sim \mathcal{N}\left(u_i^1(t)u_j^2(t), 0.05\right)$. We randomly sampled 500 (irregular) timestamps from $[0, 1]$. For each timestamp, we randomly picked two entries, and sampled their values accordingly. Overall, we sampled 1,000 observed tensor entry values.

We implemented SFTL with PyTorch [Paszke et al., 2019]. We used $\nu = \frac{3}{2}$ and $a = \rho = 0.3$ for the Matérn kernel. We streamed the sampled entries according to their timestamps, and ran our streaming factor trajectory inference based on the CP form. The estimated trajectories are shown in Fig. 2. As we can see, SFTL recovers the ground-truth pretty accurately, showing that SFTL has successfully captured the temporal evolution of the factor representation for every node. It is interesting to observe that when $t$ is around 0, 0.5 and 1, the posterior standard deviation (the shaded region) increases significantly. This is reasonable: the ground-truth trajectories overlap at these time points, making it more difficult to differentiate/estimate their values at these time points. Accordingly, the uncertainty of the estimation increases. In Section D of Appendix, we further provide the root-mean-square error (RMSE) in recovering the four trajectories, and sensitivity analysis of the kernel parameters.

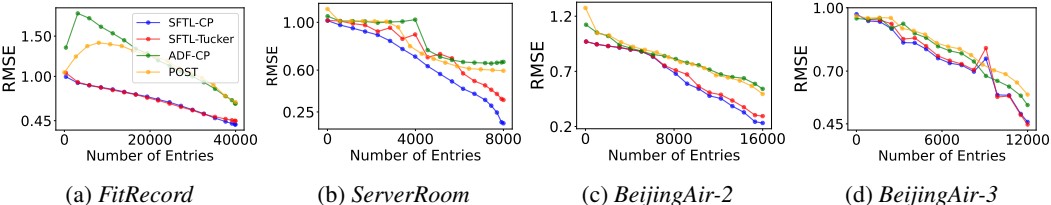

(a) *FitRecord*  (b) *ServerRoom*  (c) *BeijingAir-2*  (d) *BeijingAir-3*

Figure 3: Online prediction error with the number of processed entries ($R = 5$).

## 6.2 Real-World Applications

Next, we examined SFTL in four real-world datasets: *FitRecord*, *ServerRoom*, *BeijingAir-2*, and *BeijingAir-3*. We tested 11 competing approaches. We compared with state-of-the-art streaming tensor decomposition methods based on the CP or Tucker model, including (1) POST [Du et al., 2018], (2) BASS-Tucker [Fang et al., 2021a] and (3) ADF-CP [Wang and Zhe, 2019], the state-of-the-art static decomposition algorithms, including (4) P-Tucker [Oh et al., 2018], (5) CP-ALS and (6) Tucker-ALS [Bader and Kolda, 2008]. For those methods, we augment the tensor with a time mode, and convert the ordered, unique timestamps into increasing time steps. We also compared with the most recent continuous-time decomposition methods. (7) CT-CP [Zhang et al., 2021], (8) CT-GP, (9) BCTT [Fang et al., 2022], (10) THIS-ODE [Li et al., 2022], and (11) NONFAT [Wang et al., 2022], nonparametric factor trajectory learning, the only existing work that also estimates factor trajectories for temporal tensor decomposition. Note that the methods 4-11 cannot handle data streams. They have to iteratively access the data to update the model parameters and factor estimates. The details about the competing methods and datasets are provided in Appendix (Section E).

For all the competing methods, we used the publicly released implementations of the original authors. The hyper-parameter setting and turning follows the original papers. For SFTL, we chose $\nu$ from $\{\frac{1}{2}, \frac{3}{2}\}$, $a$ from [0.5, 1] and $\rho$ from [0.1, 0.5]. For our online filtering, the maximum number of CEP iterations was set to $50$ and the tolerance level to $10^{-4}$. For numerical stability, we re-scaled the timestamps to $[0, 1]$. We examined the number of factors (or factor trajectories) $R \in \{2, 3, 5, 7\}$.

**Final Prediction Accuracy.** We first examined the final prediction accuracy with our learned factor trajectories. To this end, we followed [Xu et al., 2012, Kang et al., 2012], and randomly sampled $80\%$ observed entry values and their timestamps for streaming inference and then tested the prediction error on the remaining entries. We also compared with the static decomposition methods, which need to repeatedly access the training entries. We repeated the experiment five times, and computed the average root mean-square-error (RMSE), average mean-absolute-error (MAE), and their standard deviations. We ran our method based on both the CP and Tucker forms, denoted by SFTL-CP and SFTL-Tucker, respectively. We report the results for $R = 5$ in Table 1. Due to the space limit, we leave the other results in the Appendix (Table 4,5, and 6). As we can see, SFTL outperforms all the streaming approaches by a large margin. SFTL even obtains significantly better prediction accuracy than all the static decomposition approaches, except that on *Server Room*, SFTL is second to THIS-ODE and NONFAT. Note that SFTL only went through the training entries for once. Although NONFAT can also estimate the factor trajectories, it uses GPs to perform black-box nonlinear decomposition and hence loses the interpretability. Note that NONFAT in most case also outperforms the other static decomposition methods that only estimate time-invariant factors. The superior performance of SFTL and NONFAT shows the importance of capturing factor evolution.

**Online Predictive Performance.** Next, we evaluated the online predictive performance of SFTL. Whenever a batch of entries at a new timestamp has been processed, we examined the prediction accuracy on the test set, with our current estimate of the factor trajectories. We repeated the evaluation for five times, and examine how the average prediction error varies along with the number of processed entries. We show the results for $R = 5$ in Fig. 3, and the others in Appendix (Fig. 5, 6, and 7). It is clear that SFTL in most cases outperforms the competing streaming decomposition algorithms by a large margin throughout the course of running. Note that the online behavior of BASS-Tucker was quite unstable and so we excluded it in the figures. It confirms the advantage of our streaming trajectory learning approach — even in the streaming scenario, incrementally capturing the time-variation of the factors can perform better than updating fixed, static factors. To confirm the advantage of SFTL in computational efficiency, we report the running time in Section G of Appendix.

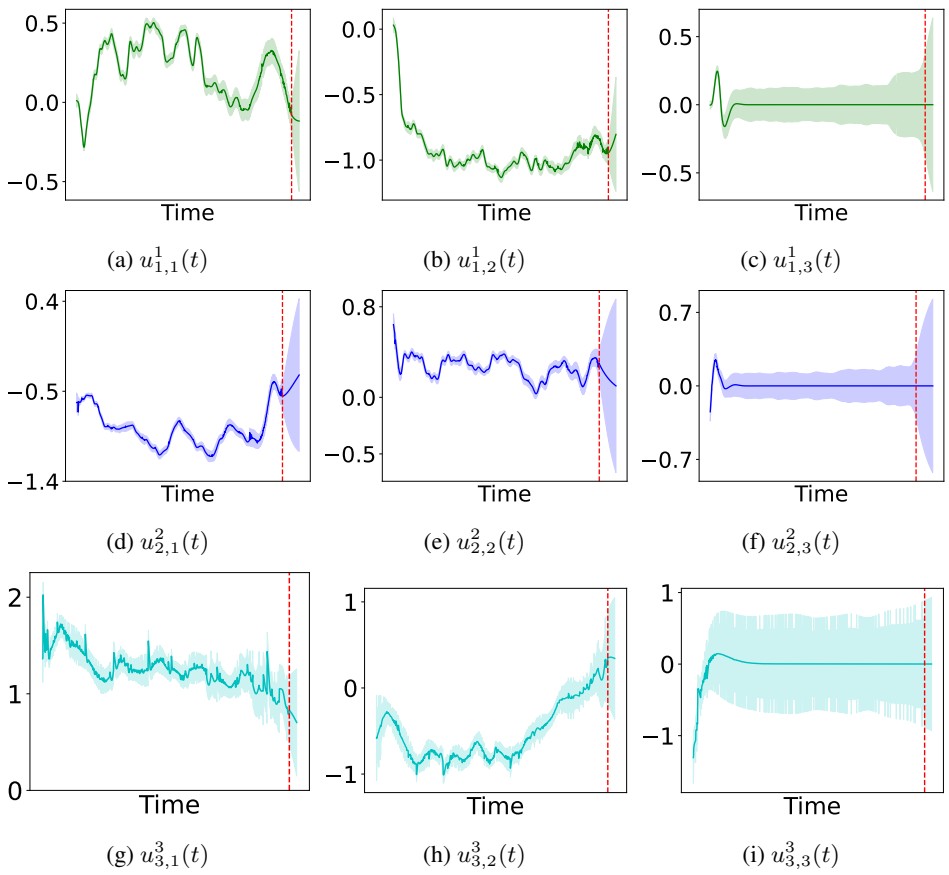

Figure 4: The learned factor trajectories of object 1, 2, 3 in mode 1, 2, 3, respectively, from *ServerRoom* data. The shaded region shows one standard deviation and the dashed line indicates the biggest timestamp in the data.

**Investigation of Learning Results.** Finally, we investigated our learned factor trajectories. We set $R = 3$ and ran SFTL-CP on *ServerRoom*. In Fig. 4, we visualize the trajectory for the 1st air conditioning mode, the 2nd power usage level, and the 3rd location. The shaded region indicates the standard deviation, and the red dashed line the biggest timestamp in the data. First, we can see that the posterior variance of all the trajectories grow quickly when moving to the right of the red line. This is reasonable because it is getting far away from the training region. Second, for each object, the first and second trajectories (1st and 2nd column in Fig. 4) exhibit quite different time-varying patterns, *e.g.,* the local periodicity in $u_{1,2}^1(t)$ and $u_{2,2}^2(t)$, the gradual decreasing and increasing trends in $u_{3,1}^3(t)$ and $u_{3,2}^3(t)$, respectively, which imply different inner properties of the object. Third, it is particularly interesting to see that the third trajectory for all the objects appear to be close to zero, with relatively large posterior variance all the time. This might imply that two factor trajectories have been sufficient to represent each object. Requesting for a third trajectory is redundant. More important, our model is empirically able to detect such redundancy and returns a zero-valued trajectory.

## 7   Conclusion

We have presented SFTL, a probabilistic temporal tensor decomposition approach. SFTL can efficiently handle streaming data, and estimate time-varying factor representations. On four real-world applications, SFTL achieves superior online and final prediction accuracy.

## Acknowledgments

This work has been supported by MURI AFOSR grant FA9550-20-1-0358 and NSF CAREER Award IIS-2046295.

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

# Appendix

## A  Spectral Analysis and LTI-SDE

We consider the Matérn kernel family,

$$\kappa_\nu(t, t') = a \frac{\left(\frac{\sqrt{2\nu}}{\rho}\Delta\right)^\nu}{\Gamma(\nu)2^{\nu-1}} K_\nu\left(\frac{\sqrt{2\nu}}{\rho}\Delta\right), \tag{10}$$

where $\Delta = |t - t'|$, $\Gamma(\cdot)$ is the Gamma function, $a > 0$ and $\rho > 0$ are the amplitude and length-scale parameters, respectively, $K_\nu$ is the modified Bessel function of the second kind, $\nu > 0$ controls the smoothness. Since $\kappa_\nu$ is a stationary kernel, *i.e.,* $\kappa_\nu(t, t') = \kappa_\nu(t - t')$, according to the Wiener-Khinchin theorem [Chatfield, 2003], if

$$f(t) \sim \mathcal{GP}(0, \kappa_\nu(t, t')),$$

the energy spectrum density of $f(t)$ can be obtained by the Fourier transform of $\kappa_\nu(\Delta)$,

$$S(\omega) = a \frac{2\sqrt{\pi}\Gamma(\frac{1}{2} + \nu)}{\Gamma(\nu)} \alpha^{2\nu}\left(\alpha^2 + \omega^2\right)^{-\left(\nu+\frac{1}{2}\right)}, \tag{11}$$

where $\omega$ is the frequency, and $\alpha = \frac{\sqrt{2\nu}}{\rho}$. We consider the commonly used choice $\nu = p + \frac{1}{2}$ where $p \in \{0, 1, 2, \ldots\}$. Then we can observe that

$$S(\omega) = \frac{\sigma^2}{(\alpha^2 + \omega^2)^{p+1}} = \frac{\sigma^2}{(\alpha + i\omega)^{p+1}(\alpha - i\omega)^{p+1}}, \tag{12}$$

where $\sigma^2 = a\frac{2\sqrt{\pi}\Gamma(p+1)}{\Gamma(p+\frac{1}{2})}\alpha^{2p+1}$, and $i$ indicates an imaginary number. We expand the polynomial

$$(\alpha + i\omega)^{p+1} = \sum_{k=0}^{p} c_k(i\omega)^k + (i\omega)^{p+1}, \tag{13}$$

where $\{c_k | 0 \le k \le p\}$ are the coefficients. From (12) and (13), we can construct an equivalent system to generate the signal $f(t)$. That is, in the frequency domain, the system output's Fourier transform $\widehat{f}(\omega)$ is given by

$$\sum_{k=1}^{p} c_k(i\omega)^k \widehat{f}(\omega) + (i\omega)^{p+1}\widehat{f}(\omega) = \widehat{\beta}(\omega), \tag{14}$$

where $\widehat{\beta}$ is the Fourier transform of a white noise process $\beta(t)$ with spectral density (or diffusion) $\sigma^2$. The reason is that by construction, $\widehat{f}(\omega) = \frac{\widehat{\beta}(\omega)}{(\alpha+i\omega)^{p+1}}$, which gives exactly the same spectral density as in (12), $S(\omega) = |\widehat{f}(\omega)|^2$. We then conduct inverse Fourier transform on both sides of (14) to obtain the representation in the time domain,

$$\sum_{k=1}^{p} c_k \frac{\mathrm{d}^k f}{\mathrm{d}t^k} + \frac{\mathrm{d}^{p+1}f}{\mathrm{d}t^{p+1}} = \beta(t), \tag{15}$$

which is an SDE. Note that $\beta(t)$ has the density $\sigma^2$. We can further construct a new state $\mathbf{z} = (f, f^{(1)}, \ldots, f^{(p)})^\top$ (where each $f^{(k)} \triangleq \mathrm{d}^k f/\mathrm{d}t^k$) and convert (15) into a linear time-invariant (LTI) SDE,

$$\frac{\mathrm{d}\mathbf{z}}{\mathrm{d}t} = \mathbf{A}\mathbf{z} + \boldsymbol{\eta} \cdot \beta(t), \tag{16}$$

where

$$\mathbf{A} = \begin{pmatrix} 0 & 1 & & \\ & \ddots & \ddots & \\ & & 0 & 1 \\ -c_0 & \ldots & -c_{p-1} & -c_p \end{pmatrix}, \quad \boldsymbol{\eta} = \begin{pmatrix} 0 \\ \vdots \\ 0 \\ 1 \end{pmatrix}.$$

For a concrete example, if we take $p = 1$ (and so $\nu = \frac{3}{2}$), then $\mathbf{A} = [0, 1; -\alpha^2, -2\alpha]$, $\boldsymbol{\eta} = [0; 1]$, and $\sigma^2 = 4a\alpha^3$.

The LTI-SDE is particularly useful in that its finite set of states follow a Gauss-Markov chain, namely the state-space prior. Specifically, given arbitrary $t_1 < \ldots < t_L$, we have

$$p(\mathbf{z}(t_1), \ldots, \mathbf{z}(t_L)) = p(\mathbf{z}(t_1)) \prod_{k=1}^{L-1} p(\mathbf{z}(t_{k+1})|\mathbf{z}(t_k)),$$

where $p(\mathbf{z}(t_1)) = \mathcal{N}(\mathbf{z}(t_1)|\mathbf{0}, \mathbf{P}_\infty)$, $p(\mathbf{z}(t_{k+1})|\mathbf{z}(t_k)) = \mathcal{N}(\mathbf{z}(t_{k+1})|\mathbf{F}_k\mathbf{z}(t_k), \mathbf{Q}_k)$, $\mathbf{P}_\infty$ is the stationary covariance matrix computed by solving the matrix Riccati equation [Lancaster and Rodman, 1995], $\mathbf{F}_n = \exp(\Delta_k \cdot \mathbf{A})$ where $\Delta_k = t_{k+1} - t_k$, and $\mathbf{Q}_k = \mathbf{P}_\infty - \mathbf{A}_k\mathbf{P}_\infty\mathbf{A}_k^\top$. Therefore, we do not need the full covariance matrix as in the standard GP prior, and the computation is much more efficient. The chain structure is also convenient to handle streaming data as explained in the main paper.

Note that for other type of kernel functions, such as the square exponential (SE) kernel, we can approximate the inverse spectral density $1/S(\omega)$ with a polynomial of $\omega^2$ with negative roots, and follow the same way to construct an LTI-SDE (approximation) and state-space prior.

# B    RTS Smoother

Consider a standard state-space model with state $\mathbf{x}_n$ and observation $\mathbf{y}_n$ at each time step $n$. The prior distribution is a Gauss-Markov chain,

$$p(\mathbf{x}_{n+1}|\mathbf{x}_n) = \mathcal{N}(\mathbf{x}_{n+1}|\mathbf{A}_n\mathbf{x}_n, \mathbf{Q}_n),$$
$$p(\mathbf{x}_0) = \mathcal{N}(\mathbf{x}_0|\mathbf{m}_0, \mathbf{P}_0).$$

Suppose we have a Gaussian observation likelihood,

$$p(\mathbf{y}_n|\mathbf{x}_n) = \mathcal{N}(\mathbf{y}_n|\mathbf{H}_n\mathbf{x}_n, \mathbf{W}_n).$$

Then upon receiving each $\mathbf{y}_n$, we can use Kalman filtering to obtain the exact running posterior,

$$p(\mathbf{x}_n|\mathbf{y}_{1:n}) = \mathcal{N}(\mathbf{x}_n|\mathbf{m}_k, \mathbf{P}_k),$$

which is a Gaussian. After all the data has been processed — suppose it ends after step $N$ — we can use Rauch–Tung–Striebel (RTS) smoother [Särkkä, 2013] to efficiently compute the full posterior of each state from backward, which does not need to re-access any data: $p(\mathbf{x}_n|\mathbf{y}_{1:N}) = \mathcal{N}(\mathbf{x}_n|\mathbf{m}_n^s, \mathbf{P}_n^s)$, where

$$\mathbf{m}_{n+1}^- = \mathbf{A}_n\mathbf{m}_n, \quad \mathbf{P}_{n+1}^- = \mathbf{A}_n\mathbf{P}_n\mathbf{A}_n^\top + \mathbf{Q}_n,$$
$$\mathbf{G}_n = \mathbf{P}_n\mathbf{A}_n^\top[\mathbf{P}_{n+1}^-]^{-1},$$
$$\mathbf{m}_n^s = \mathbf{m}_n + \mathbf{G}_n\left(\mathbf{m}_{n+1}^s - \mathbf{m}_{n+1}^-\right),$$
$$\mathbf{P}_n^s = \mathbf{P}_n + \mathbf{G}_n[\mathbf{P}_{n+1}^s - \mathbf{P}_{n+1}^-]\mathbf{G}_n^\top. \tag{17}$$

As we can see, the computation only needs the running posterior $p(\mathbf{x}_n|\mathbf{y}_{1:n}) = \mathcal{N}(\cdot|\mathbf{m}_n, \mathbf{P}_n)$ and the full posterior of the next state $p(\mathbf{x}_{n+1}|\mathbf{y}_{1:N}) = \mathcal{N}(\cdot|\mathbf{m}_{n+1}, \mathbf{P}_{n+1})$. It does not need to revisit previous observations $\mathbf{y}_{1:N}$

# C    Details about Online Trajectory Inference

In this section, we provide the details about how to update the running posterior according to (8) and (9) (in the main paper) with the conditional EP (CEP) framework [Wang and Zhe, 2019].

## C.1    EP and CEP framework

We first give a brief introduction to the EP and CEP framework. Consider a general probabilistic model with latent parameters $\boldsymbol{\theta}$. Given the observed data $\mathcal{D} = \{\mathbf{y}_1, \ldots, \mathbf{y}_N\}$, the joint probability distribution is

$$p(\boldsymbol{\theta}, \mathcal{D}) = p(\boldsymbol{\theta}) \prod_{n=1}^{N} p(\mathbf{y}_n|\boldsymbol{\theta}). \tag{18}$$

Our goal is to compute the posterior $p(\boldsymbol{\theta}|\mathcal{D})$. However, it is usually infeasible to compute the exact marginal distribution $p(\mathcal{D})$, because of the complexity of the likelihood and/or prior. EP therefore seeks to approximate each term in the joint probability by an exponential-family term,

$$p(y_n|\boldsymbol{\theta}) \approx c_n f_n(\boldsymbol{\theta}), \quad p(\boldsymbol{\theta}) \approx c_0 f_0(\boldsymbol{\theta}), \tag{19}$$

where $c_n$ and $c_0$ are constants to ensure the normalization consistency (they will get canceled in the inference, so we do not need to calculate them), and

$$f_n(\boldsymbol{\theta}) \propto \exp(\boldsymbol{\lambda}_n^\top \boldsymbol{\phi}(\boldsymbol{\theta})), (0 \leq n \leq N)$$

where $\boldsymbol{\lambda}_n$ is the natural parameter and $\boldsymbol{\phi}(\boldsymbol{\theta})$ is sufficient statistics. For example, if we choose a Gaussian term, $f_n = \mathcal{N}(\boldsymbol{\theta}|\boldsymbol{\mu}_n, \boldsymbol{\Sigma}_n)$, then the sufficient statistics is $\boldsymbol{\phi}(\boldsymbol{\theta}) = \{\boldsymbol{\theta}, \boldsymbol{\theta\theta}^\top\}$. The moment is the expectation of the sufficient statistics.

We therefore approximate the joint probability with

$$p(\boldsymbol{\theta}, \mathcal{D}) = p(\boldsymbol{\theta}) \prod_{n=1}^{N} p(\mathbf{y}_n|\boldsymbol{\theta}) \approx f_0(\boldsymbol{\theta}) \prod_{n=1}^{N} f_n(\boldsymbol{\theta}) \cdot \text{const.} \tag{20}$$

Because the exponential family is closed under product operations, we can immediately obtain a closed-form approximate posterior $q(\boldsymbol{\theta}) \approx p(\boldsymbol{\theta}|\mathcal{D})$ by merging the approximation terms in the R.H.S of (20), which is still a distribution in the exponential family.

Then the task amounts to optimizing those approximation terms $\{f_n(\boldsymbol{\theta})|0 \leq n \leq N\}$. EP repeatedly conducts four steps to optimize each $f_n$.

- **Step 1.** We obtain the calibrated distribution that integrates the context information of $f_n$,

$$q^{\backslash n}(\boldsymbol{\theta}) \propto \frac{q(\boldsymbol{\theta})}{f_n(\boldsymbol{\theta})},$$

  where $q(\boldsymbol{\theta})$ is the current posterior approximation.

- **Step 2.** We construct a tilted distribution to combine the true likelihood,

$$\widetilde{p}(\boldsymbol{\theta}) \propto q^{\backslash n}(\boldsymbol{\theta}) \cdot p(\mathbf{y}_n|\boldsymbol{\theta}).$$

  Note that if $n = 0$, we have $\widetilde{p}(\boldsymbol{\theta}) \propto q^{\backslash n}(\boldsymbol{\theta}) \cdot p(\boldsymbol{\theta})$.

- **Step 3.** We project the tilted distribution back to the exponential family,

$$q^*(\boldsymbol{\theta}) = \operatorname*{argmin}_q \ \text{KL}(\widetilde{p}\|q)$$

  where $q$ belongs to the exponential family. This can be done by moment matching,

$$\mathbb{E}_{q^*}[\boldsymbol{\phi}(\boldsymbol{\theta})] = \mathbb{E}_{\widetilde{p}}[\boldsymbol{\phi}(\boldsymbol{\theta})]. \tag{21}$$

  That is, we compute the expected moment under $\widetilde{p}$, with which to obtain the parameters of $q^*$. For example, if $q^*(\boldsymbol{\theta})$ is a Gaussian distribution, then we need to compute $\mathbb{E}_{\widetilde{p}}[\boldsymbol{\theta}]$ and $\mathbb{E}_{\widetilde{p}}[\boldsymbol{\theta\theta}^\top]$, with which to obtain the mean and covariance for $q^*(\boldsymbol{\theta})$. Hence we obtain $q^*(\boldsymbol{\theta}) = \mathcal{N}(\boldsymbol{\theta}|\mathbb{E}_{\widetilde{p}}[\boldsymbol{\theta}], \mathbb{E}_{\widetilde{p}}[\boldsymbol{\theta\theta}^\top] - \mathbb{E}_{\widetilde{p}}[\boldsymbol{\theta}]\mathbb{E}_{\widetilde{p}}[\boldsymbol{\theta}]^\top)$

- **Step 4.** We update the approximation term by

$$f_n(\boldsymbol{\theta}) \approx \frac{q^*(\boldsymbol{\theta})}{q^{\backslash}(\boldsymbol{\theta})}. \tag{22}$$

In practice, EP often updates all the $f_n$'s in parallel, and uses damping to avoid divergence. It iteratively runs the four steps until convergence. In essence, this is a fixed point iteration to optimize a free energy function (a mini-max problem) [Minka, 2001].

The critical step in EP is the moment matching (21). However, in many cases, it is analytically intractable to compute the moment under the tilted distribution $\widetilde{p}$, due to the complexity of the likelihood. To address this problem, CEP considers the commonly used case that each $f_n$ has a factorized structure,

$$f_n(\boldsymbol{\theta}) = \prod_m f_{nm}(\boldsymbol{\theta}_m), \tag{23}$$

where each $f_{nm}$ is also in the exponential family, and $\{\boldsymbol{\theta}_m\}$ are mutually disjoint. Then at the moment matching step, we need to compute the moment of each $\boldsymbol{\theta}_m$ under $\widetilde{p}$, *i.e.,* $\mathbb{E}_{\widetilde{p}}[\boldsymbol{\phi}(\boldsymbol{\theta}_m)]$. The first key idea of CEP is to use the nested structure,

$$\mathbb{E}_{\widetilde{p}}[\boldsymbol{\phi}(\boldsymbol{\theta}_m)] = \mathbb{E}_{\widetilde{p}(\boldsymbol{\theta}_{\backslash m})}\mathbb{E}_{\widetilde{p}(\boldsymbol{\theta}_m|\boldsymbol{\theta}_{\backslash m})}[\boldsymbol{\phi}(\boldsymbol{\theta}_m)], \tag{24}$$

where $\boldsymbol{\theta}_{\backslash m} = \boldsymbol{\theta}\backslash\boldsymbol{\theta}_m$. Therefore, we can first compute the inner expectation, *i.e.,* conditional moment,

$$\mathbb{E}_{\widetilde{p}(\boldsymbol{\theta}_m|\boldsymbol{\theta}_{\backslash m})}[\boldsymbol{\phi}(\boldsymbol{\theta}_m)] = \mathbf{g}(\boldsymbol{\theta}_{\backslash m}), \tag{25}$$

and then seek for computing the outer expectation, $\mathbb{E}_{\widetilde{p}(\boldsymbol{\theta}_{\backslash m})}[\mathbf{g}(\boldsymbol{\theta}_{\backslash m})]$. The inner expectation is often easy to compute (*e.g.,* with our CP/Tucker likelihood). When $f_n$ is factorized individually over each element of $\boldsymbol{\theta}$, this can always be efficiently and accurately calculated by quadrature. However, the outer expectation is still difficult to obtain because $\widetilde{p}(\boldsymbol{\theta}_{\backslash m})$ is intractable. The second key idea of CEP is that since the moment matching is also between $q(\boldsymbol{\theta}_{\backslash m})$ and $\widetilde{p}(\boldsymbol{\theta}_{\backslash m})$, we can use the current marginal posterior to approximate the marginal titled distribution and then compute the outer expectation,

$$\mathbb{E}_{\widetilde{p}(\boldsymbol{\theta}_{\backslash m})}[\mathbf{g}(\boldsymbol{\theta}_{\backslash m})] \approx \mathbb{E}_{q(\boldsymbol{\theta}_{\backslash m})}[\mathbf{g}(\boldsymbol{\theta}_{\backslash m})]. \tag{26}$$

If it is still analytically intractable, we can use the delta method [Oehlert, 1992] to approximate the expectation. That is, we use a Taylor expansion of $\mathbf{g}(\cdot)$ at the mean of $\boldsymbol{\theta}_{\backslash m}$. Take the first-order expansion as an example,

$$\mathbf{g}(\boldsymbol{\theta}_{\backslash m}) \approx \mathbf{g}\left(\mathbb{E}_{q(\boldsymbol{\theta}_{\backslash m})}[\boldsymbol{\theta}_{\backslash m}]\right) + \mathbf{J}\left(\boldsymbol{\theta}_{\backslash m} - \mathbb{E}_{q(\boldsymbol{\theta}_{\backslash m})}[\boldsymbol{\theta}_{\backslash m}]\right)$$

where $\mathbf{J}$ is the Jacobian of $\mathbf{g}$ at $\mathbb{E}_{q(\boldsymbol{\theta}_{\backslash m})}[\boldsymbol{\theta}_{\backslash m}]$. Then we take the expectation on the Taylor approximation instead,

$$\mathbb{E}_{q(\boldsymbol{\theta}_{\backslash m})}\left[\mathbf{g}(\boldsymbol{\theta}_{\backslash m})\right] \approx \mathbf{g}\left(\mathbb{E}_{q(\boldsymbol{\theta}_{\backslash m})}[\boldsymbol{\theta}_{\backslash m}]\right). \tag{27}$$

The above computation is very convenient to implement. Once we obtain the conditional moment $\mathbf{g}(\boldsymbol{\theta}_{\backslash m})$, we simply replace the $\boldsymbol{\theta}_{\backslash m}$ by its expectation under current posterior approximation $q$, *i.e.,* $\mathbb{E}_{q(\boldsymbol{\theta}_{\backslash m})}[\boldsymbol{\theta}_{\backslash m}]$, to obtain the matched moment $\mathbf{g}(\mathbb{E}_{q(\boldsymbol{\theta}_{\backslash m})}[\boldsymbol{\theta}_{\backslash m}])$, with which to construct $q^*$ in Step 3 of EP (see (21)). The remaining steps are the same.

## C.2 Running Posterior Update

Now we use the CEP framework to update the running posterior $p(\Theta_{n+1}, \tau|\mathcal{D}_{t_{n+1}})$ in (8) via the approximation (9). To simplify the notation, let us define $\mathbf{v}_{l_m}^m \triangleq \mathbf{u}_{\ell_m}^m(t_{n+1})$, and hence for each $(\boldsymbol{\ell}, y) \in \mathcal{B}_{n+1}$, we approximate

$$\mathcal{N}\left(y|\mathbf{1}^\top\left(\mathbf{v}_{\ell_1}^1 \circ \ldots \circ \mathbf{v}_{\ell_M}^M\right), \tau^{-1}\right) \approx \prod_{m=1}^M \mathcal{N}(\mathbf{v}_{\ell_m}^m|\boldsymbol{\gamma}_{\ell_m}^m, \boldsymbol{\Sigma}_{\ell_m}^m)\text{Gam}(\tau|\alpha_{\boldsymbol{\ell}}, \omega_{\boldsymbol{\ell}}). \tag{28}$$

If we substitute (9) into (8), we can immediately obtain a Gaussian posterior approximation of each $\mathbf{v}_{\ell_m}^m$ and a Gamma posterior approximation of the noise inverse variance $\tau$. Then dividing the current posterior approximation with the R.H.S of (28), we can obtain the calibrated distribution,

$$q^{\backslash\boldsymbol{\ell}}(\mathbf{v}_{\ell_m}^m) = \mathcal{N}(\mathbf{v}_{\ell_m}^m|\boldsymbol{\beta}_{\ell_m}^m, \boldsymbol{\Omega}_{\ell_m}^m),$$
$$q^{\backslash\boldsymbol{\ell}}(\tau) = \text{Gam}(\tau|\alpha^{\backslash\boldsymbol{\ell}}, \omega^{\backslash\boldsymbol{\ell}}), \tag{29}$$

where $1 \leq m \leq M$. Next, we construct a tilted distribution,

$$\widetilde{p}(\mathbf{v}_{\ell_1}^1, \ldots, \mathbf{v}_{\ell_M}^M, \tau) \propto q^{\backslash\boldsymbol{\ell}}(\tau) \cdot \prod_{m=1}^M q^{\backslash\boldsymbol{\ell}}(\mathbf{v}_{\ell_m}^m) \cdot \mathcal{N}\left(y|\mathbf{1}^\top\left(\mathbf{v}_{\ell_1}^1 \circ \ldots \circ \mathbf{v}_{\ell_M}^M\right), \tau^{-1}\right). \tag{30}$$

To update each $\mathcal{N}(\mathbf{v}_{\ell_m}^m|\boldsymbol{\gamma}_{\ell_m}^m, \boldsymbol{\Sigma}_{\ell_m}^m)$ in (28), we first look into the conditional tilted distribution,

$$\widetilde{p}(\mathbf{v}_{\ell_m}^m|\mathcal{V}_{\boldsymbol{\ell}}^{\backslash m}, \tau) \propto \mathcal{N}(\mathbf{v}_{\ell_m}^m|\boldsymbol{\beta}_{\ell_m}^m, \boldsymbol{\Omega}_{\ell_m}^m) \cdot \mathcal{N}\left(y|\left(\mathbf{v}_{\ell_m}^m\right)^\top \mathbf{v}_{\boldsymbol{\ell}}^{\backslash m}, \tau^{-1}\right) \tag{31}$$

where $\mathcal{V}_{\boldsymbol{\ell}}^{\backslash m}$ is $\{\mathbf{v}_{\ell_j}^j | 1 \leq j \leq M, j \neq m\}$, and

$$\mathbf{v}_{\boldsymbol{\ell}}^{\backslash m} = \mathbf{v}_{\ell_1}^1 \circ \ldots \circ \mathbf{v}_{\ell_{m-1}}^{m-1} \circ \mathbf{v}_{\ell_{m+1}}^{m+1} \circ \ldots \circ \mathbf{v}_{\ell_M}^M.$$

The conditional tilted distribution is obviously Gaussian, and the conditional moment is straightforward to obtain,

$$\mathbf{S}(\mathbf{v}_{\ell_m}^m | \mathcal{V}_{\boldsymbol{\ell}}^{\backslash m}, \tau) = \left[ \boldsymbol{\Omega}_{\ell_m}^{m\,-1} + \tau \mathbf{v}_{\boldsymbol{\ell}}^{\backslash m} \left( \mathbf{v}_{\boldsymbol{\ell}}^{\backslash m} \right)^\top \right]^{-1}, \tag{32}$$

$$\mathbb{E}[\mathbf{v}_{\ell_m}^m | \mathcal{V}_{\boldsymbol{\ell}}^{\backslash m}, \tau] = \mathbf{S}(\mathbf{v}_{\ell_m}^m | \mathcal{V}_{\boldsymbol{\ell}}^{\backslash m}, \tau) \cdot \left( \boldsymbol{\Omega}_{\ell_m}^{m\,-1} \boldsymbol{\beta}_{\ell_m}^m + \tau y \mathbf{v}_{\boldsymbol{\ell}}^{\backslash m} \right), \tag{33}$$

where $\mathbf{S}$ denotes the conditional covariance. Next, according to (27), we simply replace $\tau$, $\mathbf{v}_{\boldsymbol{\ell}}^{\backslash m}$, and $\mathbf{v}_{\boldsymbol{\ell}}^{\backslash m} \left( \mathbf{v}_{\boldsymbol{\ell}}^{\backslash m} \right)^\top$ by their expectation under the current posterior $q$ in (32) and (33), to obtain the moments, *i.e.*, the mean and covariance matrix, with which we can construct $q^*$ in Step 3 of the EP framework. The computation of $\mathbb{E}_q[\tau]$ is straightforward, and

$$\mathbb{E}_q[\mathbf{v}_{\boldsymbol{\ell}}^{\backslash m}] = \mathbb{E}_q[\mathbf{v}_{\ell_1}^1] \circ \ldots \circ \mathbb{E}_q[\mathbf{v}_{\ell_{m-1}}^{m-1}] \circ \mathbb{E}_q[\mathbf{v}_{\ell_{m+1}}^{m+1}] \circ \ldots \circ \mathbb{E}_q[\mathbf{v}_{\ell_M}^M],$$

$$\mathbb{E}_q[\mathbf{v}_{\boldsymbol{\ell}}^{\backslash m} \left( \mathbf{v}_{\boldsymbol{\ell}}^{\backslash m} \right)^\top] = \mathbb{E}_q[\mathbf{v}_{\ell_1}^1 \left( \mathbf{v}_{\ell_1}^1 \right)^\top] \circ \ldots \circ \mathbb{E}_q[\mathbf{v}_{\ell_{m-1}}^{m-1} \left( \mathbf{v}_{\ell_{m-1}}^{m-1} \right)^\top]$$

$$\circ \mathbb{E}_q[\mathbf{v}_{\ell_{m+1}}^{m+1} \left( \mathbf{v}_{\ell_{m+1}}^{m+1} \right)^\top] \circ \ldots \circ \mathbb{E}_q[\mathbf{v}_{\ell_M}^M \left( \mathbf{v}_{\ell_M}^M \right)^\top].$$

Similarly, to update $\text{Gam}(\alpha_{\boldsymbol{\ell}}, \omega_{\boldsymbol{\ell}})$ in (28), we first observe that the conditional titled distribution is also a Gamma distribution,

$$\widetilde{p}(\tau | \mathcal{V}_{\boldsymbol{\ell}}) \propto \text{Gam}(\tau | \widetilde{\alpha}, \widetilde{\omega}) \propto \text{Gam}(\tau | \alpha^{\backslash \boldsymbol{\ell}}, \omega^{\backslash \boldsymbol{\ell}}) \mathcal{N}(y | \mathbf{1}^\top \mathbf{v}_{\boldsymbol{\ell}}, \tau^{-1}), \tag{34}$$

where $\mathbf{v}_{\boldsymbol{\ell}} = \mathbf{v}_{\ell_1}^1 \circ \ldots \circ \mathbf{v}_{\ell_M}^M$, and

$$\widetilde{\alpha} = \alpha^{\backslash \boldsymbol{\ell}} + \frac{1}{2},$$

$$\widetilde{\omega} = \omega^{\backslash \boldsymbol{\ell}} + \frac{1}{2}y^2 + \frac{1}{2}\mathbf{1}^\top \mathbf{v}_{\boldsymbol{\ell}} \mathbf{v}_{\boldsymbol{\ell}}^\top \mathbf{1} - y\mathbf{1}^\top \mathbf{v}. \tag{35}$$

Since the conditional moments (the expectation of $\tau$ and $\log \tau$) are functions of $\alpha$ and $\omega$, when using the delta method to approximate the expected conditional moment, it is equivalent to approximating the expectation of $\widetilde{\alpha}$ and $\widetilde{\omega}$ first, and then use the expected $\widetilde{\alpha}$ and $\widetilde{\omega}$ to recover the moments. As a result, we can simply replace $\mathbf{v}_{\boldsymbol{\ell}}$ and $\mathbf{v}_{\boldsymbol{\ell}} \mathbf{v}_{\boldsymbol{\ell}}^\top$ in (35) by their expectation under the current posterior, and we obtain the approximation of $\mathbb{E}_q[\widetilde{\alpha}]$ and $\mathbb{E}_q[\widetilde{\omega}]$. With these approximated expectation, we then construct $q^*(\tau) = \text{Gam}(\tau | \mathbb{E}_q[\alpha], \mathbb{E}_q[\omega])$ at Step 3 in EP. The remaining steps are straightforward. The running posterior update with the Tucker form likelihood follows a similar way.

## D    More Results on Simulation Study

### D.1    Accuracy of Trajectory Recovery

We provide the quantitative result in recovering the factor trajectories. Note that there is only one competing method, NONFAT, which can also estimate factor trajectories. We therefore ran our method and NONFAT on the synthetic dataset. We then randomly sampled 500 time points in the domain and evaluate the RMSE of the learned factor trajectories for each method. As shown in Table 2, the RMSE of NONFAT on recovering $u_1^1(t)$ and $u_1^2(t)$ is close to SFTL, showing NONFAT achieved the same (or very close) quality in recovering these two trajectories. However, on $u_2^1(t)$ and $u_2^2(t)$, the RMSE of NONFAT is much larger, showing that NONFAT have failed to capture the other two trajectories. By contrast, SFTL consistently well recovered them.

### D.2    Sensitive Analysis on Kernel Parameters

To examine the sensitivity to the kernel parameters, we used the synthetic dataset, and randomly sampled 100 entries and new timestamps for evaluation. We then examined the length-scale $\rho$ and

|        | $u_1^1(t)$ | $u_2^1(t)$ | $u_1^2(t)$ | $u_2^2(t)$ |
|--------|-----------|-----------|-----------|-----------|
| SFTL   | 0.073     | 0.082     | 0.103     | 0.054     |
| NONFAT | 0.085     | 0.442     | 0.096     | 0.443     |

Table 2: RMSE in recovering trajectories on the simulation data.

| | $\rho$ | 0.1 | 0.3 | 0.5 | 0.7 | 0.9 |
|---|---|---|---|---|---|---|
| Matérn-1/2 | SFTL-CP | 0.091 | 0.064 | 0.059 | 0.056 | 0.057 |
| | SFTL-Tucker | 0.060 | 0.055 | 0.056 | 0.056 | 0.057 |
| Matérn-3/2 | SFTL-CP | 0.062 | 0.061 | 0.074 | 0.093 | 0.112 |
| | SFTL-Tucker | 0.061 | 0.059 | 0.078 | 0.101 | 0.129 |

(a) Prediction RMSE with $a = 0.3$ and varying $\rho$.

| | $a$ | 0.1 | 0.3 | 0.5 | 0.7 | 0.9 |
|---|---|---|---|---|---|---|
| Matérn-1/2 | SFTL-CP | 0.056 | 0.064 | 0.057 | 0.059 | 0.063 |
| | SFTL-Tucker | 0.065 | 0.055 | 0.054 | 0.055 | 0.055 |
| Matérn-3/2 | SFTL-CP | 0.072 | 0.061 | 0.063 | 0.060 | 0.059 |
| | SFTL-Tucker | 0.098 | 0.059 | 0.064 | 0.062 | 0.061 |

(b) Prediction RMSE with $\rho = 0.3$ and varying $a$.

Table 3: Sensitive analysis of amplitude $a$ and length-scale $\rho$ on synthetic data.

amplitude $a$, for two commonly-used Matérn kernels: Matérn-1/2 and Matérn-3/2. The study was performed on SFTL based on both the CP and Tucker forms. The results are reported in Table 3. Overall, the predictive performance of SFTL is less sensitive to the amplitude parameter $a$ than to the length-scale parameter $\rho$. But when we use Matérn-1/2, the performance of both SFTL-CP and SFTL-Tucker is quite stable to the length-scale parameter $\rho$. When we use Matérn-3/2, the choice of the length-scale is critical.

# E   Real-World Dataset Information and Competing Methods

We tested all the methods in the following four real-world datasets.

- *FitRecord*[3], workout logs of EndoMondo users' health status in outdoor exercises. We extracted a three-mode tensor among 500 users, 20 sports types, and 50 altitudes. The entry values are heart rates. There are 50K observed entry values along with the timestamps.

- *ServerRoom*[4], temperature logs of Poznan Supercomputing and Networking Center. We extracted a three-mode tensor between 3 air conditioning modes (24°, 27° and 30°), 3 power usage levels (50%, 75%, 100%) and 34 locations. We collected 10K entry values and their timestamps.

- *BeijingAir-2*[5], air pollution measurement in Beijing from year 2014 to 2017. We extracted a two-mode tensor (monitoring site, pollutant), of size $12 \times 6$, and collected 20K observed entry values (concentration) and their timestamps.

- *BeijingAir-3*, extracted from the same data source as *BeijingAir-2*, a three-mode tensor among 12 monitoring sites, 12 wind speeds and 6 wind directions. The entry value is the PM2.5 concentration. There are 15K observed entry values at different timestamps.

We first compared with the following state-of-the-art streaming tensor decomposition methods based on the CP or Tucker model. (1) POST [Du et al., 2018], probabilistic streaming CP decomposition via mean-field streaming variational Bayes [Broderick et al., 2013] (2) BASS-Tucker [Fang et al., 2021a] Bayesian streaming Tucker decomposition, which online estimates a sparse tensor-core via a spike-and-slab prior to enhance the interpretability. We also implemented (3) ADF-CP, streaming CP

---

[3]https://sites.google.com/eng.ucsd.edu/fitrec-project/home
[4]https://zenodo.org/record/3610078#%23.Y8SYt3bMJGi
[5]https://archive.ics.uci.edu/ml/datasets/Beijing+Multi-Site+Air-Quality+Data

decomposition by combining the assumed density filtering and conditional moment matching [Wang and Zhe, 2019].

Next, we tested the state-of-the-art static decomposition algorithms, which have to go through the data many times. (4) P-Tucker [Oh et al., 2018], an efficient Tucker decomposition algorithm that performs parallel row-wise updates. (5) CP-ALS and (6) Tucker-ALS [Bader and Kolda, 2008], CP/Tucker decomposition via alternating least square (ALS) updates. The methods (1-6) are not specifically designed for temporal decomposition and cannot utilize the timestamps of the observed entries. In order to incorporate the time information for a fair comparison, we augment the tensor with a time mode, and convert the ordered, unique timestamps into increasing time steps.

We then compared with the most recent continuous-time temporal decomposition methods. Note that none of these methods can handle data streams. They have to iteratively access the data to update the model parameters and factor estimates. (7) CT-CP [Zhang et al., 2021], continuous-time CP decomposition, which uses polynomial splines to model a time-varying coefficient $\boldsymbol{\lambda}$ for each latent factor, (8) CT-GP, continuous-time GP decomposition, which extends [Zhe et al., 2016a] to use GPs to learn the tensor entry value as a function of the latent factors and time $y_{\boldsymbol{\ell}}(t) = g(\mathbf{u}_{\ell_1}^1, \ldots, \mathbf{u}_{\ell_K}^K, t) \sim \mathcal{GP}(0, \kappa(\cdot, \cdot))$, (9) BCTT [Fang et al., 2022], Bayesian continuous-time Tucker decomposition, which estimates the tensor-core as a time-varying function, (10) THIS-ODE [Li et al., 2022], which uses a neural ODE [Chen et al., 2018] to model the entry value as a function of the latent factors and time, $\frac{\mathrm{d}y_{\boldsymbol{\ell}}(t)}{\mathrm{d}t} = \mathrm{NN}(\mathbf{u}_{\ell_1}^1, \ldots, \mathbf{u}_{\ell_K}^K, t)$ where NN is short for neural networks. (11) NONFAT [Wang et al., 2022], nonparametric factor trajectory learning, the only existing work that also estimates factor trajectories for temporal tensor decomposition. It uses a bi-level GP to estimate the trajectories in the frequency domain and applies inverse Fourier transform to return to the time domain.

## F   More Results about Prediction Accuracy

We report for $R = 2$, $R = 3$ and $R = 7$, the final prediction error (after the data has been processed) of all the methods in Table 4, Table 5, and Table 6, respectively. We report for $R = 2$, $R = 3$ and $R = 7$, the online predictive performance of the streaming decomposition approaches in Fig. 5, Fig. 6, and Fig. 7, respectively.

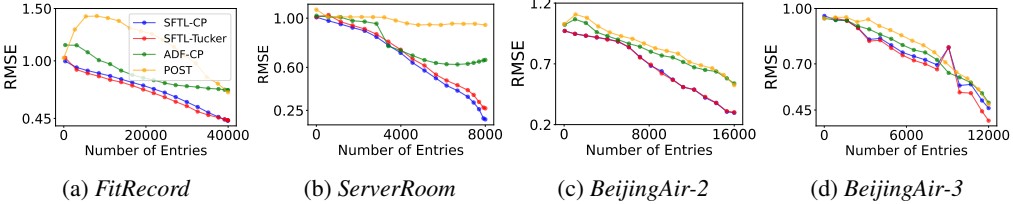

(a) *FitRecord*   (b) *ServerRoom*   (c) *BeijingAir-2*   (d) *BeijingAir-3*

Figure 5: Online prediction error with the number of processed entries ($R = 2$)

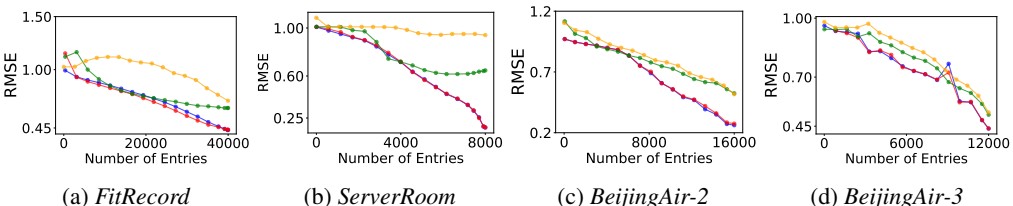

(a) *FitRecord*   (b) *ServerRoom*   (c) *BeijingAir-2*   (d) *BeijingAir-3*

Figure 6: Online prediction error with the number of processed entries ($R = 3$)

## G   Running Time

As compared with static (non-streaming) methods, such as BCTT, our method is faster and more efficient. That is because whenever new data comes in, the static methods have to retrain the model from scratch and iteratively access the whole data accumulated so far, while our method only performs

| | RMSE | *FitRecord* | *ServerRoom* | *BeijingAir-2* | *BeijingAir-3* |
|---|---|---|---|---|---|
| | PTucker | $0.606 \pm 0.015$ | $0.757 \pm 0.36$ | $0.509 \pm 0.01$ | $0.442 \pm 0.142$ |
| | Tucker-ALS | $0.914 \pm 0.01$ | $0.991 \pm 0.016$ | $0.586 \pm 0.016$ | $0.896 \pm 0.032$ |
| | CP-ALS | $0.926 \pm 0.013$ | $0.997 \pm 0.016$ | $0.647 \pm 0.041$ | $0.918 \pm 0.031$ |
| Static | CT-CP | $0.675 \pm 0.009$ | $0.412 \pm 0.024$ | $0.642 \pm 0.007$ | $0.832 \pm 0.035$ |
| | CT-GP | $0.611 \pm 0.009$ | $0.218 \pm 0.021$ | $0.723 \pm 0.01$ | $0.88 \pm 0.026$ |
| | BCTT | $0.604 \pm 0.019$ | $0.715 \pm 0.352$ | $0.504 \pm 0.01$ | $0.799 \pm 0.027$ |
| | NONFAT | $0.543 \pm 0.002$ | $\mathbf{0.132 \pm 0.002}$ | $0.425 \pm 0.002$ | $0.878 \pm 0.014$ |
| | THIS-ODE | $0.544 \pm 0.005$ | $0.142 \pm 0.004$ | $0.553 \pm 0.015$ | $0.876 \pm 0.027$ |
| | POST | $0.705 \pm 0.013$ | $0.767 \pm 0.155$ | $0.539 \pm 0.01$ | $0.695 \pm 0.135$ |
| | ADF-CP | $0.669 \pm 0.033$ | $0.764 \pm 0.114$ | $0.583 \pm 0.07$ | $0.54 \pm 0.045$ |
| Stream | BASS-Tucker | $1 \pm 0.016$ | $1 \pm 0.016$ | $1.043 \pm 0.05$ | $0.982 \pm 0.058$ |
| | SFTL-CP | $\mathbf{0.437 \pm 0.014}$ | $0.18 \pm 0.019$ | $\mathbf{0.323 \pm 0.019}$ | $0.462 \pm 0.009$ |
| | SFTL-Tucker | $0.446 \pm 0.024$ | $0.276 \pm 0.031$ | $0.344 \pm 0.031$ | $\mathbf{0.417 \pm 0.035}$ |
| | MAE | | | | |
| | PTucker | $0.416 \pm 0.005$ | $0.388 \pm 0.152$ | $0.336 \pm 0.004$ | $0.271 \pm 0.053$ |
| | Tucker-ALS | $0.676 \pm 0.008$ | $0.744 \pm 0.01$ | $0.408 \pm 0.008$ | $0.669 \pm 0.02$ |
| | CP-ALS | $0.686 \pm 0.011$ | $0.748 \pm 0.009$ | $0.454 \pm 0.057$ | $0.691 \pm 0.016$ |
| Static | CT-CP | $0.466 \pm 0.005$ | $0.295 \pm 0.029$ | $0.49 \pm 0.006$ | $0.642 \pm 0.02$ |
| | CT-GP | $0.424 \pm 0.006$ | $0.155 \pm 0.012$ | $0.517 \pm 0.01$ | $0.626 \pm 0.01$ |
| | BCTT | $0.419 \pm 0.015$ | $0.534 \pm 0.263$ | $0.343 \pm 0.003$ | $0.579 \pm 0.018$ |
| | NONFAT | $0.373 \pm 0.001$ | $\mathbf{0.083 \pm 0.001}$ | $0.282 \pm 0.002$ | $0.622 \pm 0.006$ |
| | THIS-ODE | $0.377 \pm 0.003$ | $0.097 \pm 0.003$ | $0.355 \pm 0.008$ | $0.606 \pm 0.015$ |
| | POST | $0.485 \pm 0.008$ | $0.564 \pm 0.091$ | $0.368 \pm 0.008$ | $0.517 \pm 0.123$ |
| | ADF-CP | $0.462 \pm 0.022$ | $0.574 \pm 0.073$ | $0.401 \pm 0.029$ | $0.415 \pm 0.038$ |
| Stream | BASS | $0.777 \pm 0.039$ | $0.749 \pm 0.01$ | $0.871 \pm 0.125$ | $0.727 \pm 0.029$ |
| | SFTL-CP | $\mathbf{0.248 \pm 0.005}$ | $0.126 \pm 0.007$ | $\mathbf{0.199 \pm 0.005}$ | $0.311 \pm 0.004$ |
| | SFTL-Tucker | $0.25 \pm 0.01$ | $0.203 \pm 0.032$ | $0.218 \pm 0.02$ | $\mathbf{0.261 \pm 0.023}$ |

Table 4: Final prediction error with $R = 2$. The results were averaged from five runs.

| | RMSE | *FitRecord* | *ServerRoom* | *BeijingAir-2* | *BeijingAir-3* |
|---|---|---|---|---|---|
| | PTucker | $0.603 \pm 0.045$ | $0.677 \pm 0.129$ | $0.464 \pm 0.012$ | $0.421 \pm 0.074$ |
| | Tucker-ALS | $0.885 \pm 0.007$ | $0.989 \pm 0.014$ | $0.559 \pm 0.017$ | $0.863 \pm 0.032$ |
| | CP-ALS | $0.907 \pm 0.015$ | $0.993 \pm 0.014$ | $0.594 \pm 0.031$ | $0.901 \pm 0.03$ |
| Static | CT-CP | $0.666 \pm 0.008$ | $0.5 \pm 0.2$ | $0.641 \pm 0.006$ | $0.819 \pm 0.019$ |
| | CT-GP | $0.606 \pm 0.008$ | $0.217 \pm 0.025$ | $0.749 \pm 0.014$ | $0.895 \pm 0.054$ |
| | BCTT | $0.576 \pm 0.015$ | $0.358 \pm 0.082$ | $0.454 \pm 0.011$ | $0.829 \pm 0.028$ |
| | NONFAT | $0.517 \pm 0.002$ | $\mathbf{0.129 \pm 0.002}$ | $0.408 \pm 0.005$ | $0.877 \pm 0.014$ |
| | THIS-ODE | $0.528 \pm 0.005$ | $0.132 \pm 0.002$ | $0.544 \pm 0.014$ | $0.878 \pm 0.026$ |
| | POST | $0.706 \pm 0.034$ | $0.741 \pm 0.161$ | $0.518 \pm 0.016$ | $0.622 \pm 0.123$ |
| | ADF-CP | $0.641 \pm 0.009$ | $0.652 \pm 0.012$ | $0.542 \pm 0.012$ | $0.518 \pm 0.003$ |
| Stream | BASS-Tucker | $1.008 \pm 0.017$ | $1 \pm 0.016$ | $1.035 \pm 0.038$ | $0.99 \pm 0.034$ |
| | SFTL-CP | $0.434 \pm 0.014$ | $0.178 \pm 0.006$ | $\mathbf{0.288 \pm 0.017}$ | $0.454 \pm 0.011$ |
| | SFTL-Tucker | $\mathbf{0.418 \pm 0.01}$ | $0.289 \pm 0.096$ | $0.314 \pm 0.049$ | $\mathbf{0.41 \pm 0.013}$ |
| | MAE | | | | |
| | PTucker | $0.392 \pm 0.009$ | $0.323 \pm 0.053$ | $0.307 \pm 0.005$ | $0.197 \pm 0.029$ |
| | Tucker-ALS | $0.648 \pm 0.012$ | $0.743 \pm 0.008$ | $0.39 \pm 0.008$ | $0.651 \pm 0.018$ |
| | CP-ALS | $0.666 \pm 0.013$ | $0.746 \pm 0.01$ | $0.415 \pm 0.022$ | $0.676 \pm 0.021$ |
| Static | CT-CP | $0.462 \pm 0.005$ | $0.348 \pm 0.141$ | $0.489 \pm 0.006$ | $0.632 \pm 0.015$ |
| | CT-GP | $0.419 \pm 0.005$ | $0.158 \pm 0.022$ | $0.544 \pm 0.012$ | $0.627 \pm 0.015$ |
| | BCTT | $0.392 \pm 0.004$ | $0.267 \pm 0.067$ | $0.299 \pm 0.006$ | $0.607 \pm 0.027$ |
| | NONFAT | $0.355 \pm 0.001$ | $\mathbf{0.078 \pm 0.001}$ | $0.265 \pm 0.003$ | $0.622 \pm 0.006$ |
| | THIS-ODE | $0.363 \pm 0.004$ | $0.083 \pm 0.002$ | $0.348 \pm 0.006$ | $0.603 \pm 0.009$ |
| | POST | $0.482 \pm 0.022$ | $0.54 \pm 0.102$ | $0.351 \pm 0.009$ | $0.442 \pm 0.109$ |
| | ADF-CP | $0.445 \pm 0.006$ | $0.5 \pm 0.009$ | $0.381 \pm 0.006$ | $0.393 \pm 0.009$ |
| Stream | BASS | $0.822 \pm 0.024$ | $0.749 \pm 0.009$ | $0.919 \pm 0.041$ | $0.73 \pm 0.018$ |
| | SFTL-CP | $0.246 \pm 0.005$ | $0.121 \pm 0.003$ | $\mathbf{0.176 \pm 0.006}$ | $0.305 \pm 0.006$ |
| | SFTL-Tucker | $\mathbf{0.24 \pm 0.002}$ | $0.18 \pm 0.042$ | $0.196 \pm 0.03$ | $\mathbf{0.263 \pm 0.011}$ |

Table 5: Final prediction error with $R = 3$. The results were averaged from five runs.

| | RMSE | *FitRecord* | *ServerRoom* | *BeijingAir-2* | *BeijingAir-3* |
|---|---|---|---|---|---|
| | PTucker | $0.603 \pm 0.045$ | $0.677 \pm 0.129$ | $0.464 \pm 0.012$ | $0.421 \pm 0.074$ |
| | Tucker-ALS | $0.826 \pm 0.003$ | $0.983 \pm 0.016$ | $0.586 \pm 0.018$ | $0.825 \pm 0.026$ |
| | CP-ALS | $0.878 \pm 0.012$ | $0.994 \pm 0.013$ | $0.897 \pm 0.215$ | $0.863 \pm 0.024$ |
| Static | CT-CP | $0.663 \pm 0.008$ | $0.384 \pm 0.008$ | $0.64 \pm 0.007$ | $0.818 \pm 0.019$ |
| | CT-GP | $0.603 \pm 0.006$ | $0.381 \pm 0.303$ | $0.766 \pm 0.016$ | $0.904 \pm 0.046$ |
| | BCTT | $0.498 \pm 0.011$ | $0.194 \pm 0.017$ | $0.368 \pm 0.01$ | $0.813 \pm 0.028$ |
| | NONFAT | $0.497 \pm 0.003$ | $\mathbf{0.128 \pm 0.002}$ | $0.394 \pm 0.004$ | $0.88 \pm 0.013$ |
| | THIS-ODE | $0.138 \pm 0.003$ | $0.554 \pm 0.016$ | $0.878 \pm 0.027$ | |
| | POST | $0.675 \pm 0.012$ | $0.707 \pm 0.14$ | $0.519 \pm 0.017$ | $0.738 \pm 0.068$ |
| | ADF-CP | $0.652 \pm 0.01$ | $0.646 \pm 0.008$ | $0.548 \pm 0.012$ | $0.552 \pm 0.026$ |
| Stream | BASS-Tucker | $0.604 \pm 0.043$ | $0.493 \pm 0.071$ | $0.391 \pm 0.005$ | $0.634 \pm 0.083$ |
| | SFTL-CP | $\mathbf{0.424 \pm 0.006}$ | $0.166 \pm 0.013$ | $0.256 \pm 0.013$ | $0.481 \pm 0.006$ |
| | SFTL-Tucker | $0.448 \pm 0.009$ | $0.406 \pm 0.052$ | $\mathbf{0.249 \pm 0.017}$ | $\mathbf{0.432 \pm 0.019}$ |
| | MAE | | | | |
| | PTucker | $0.353 \pm 0.005$ | $0.305 \pm 0.042$ | $0.248 \pm 0.004$ | $0.32 \pm 0.038$ |
| | Tucker-ALS | $0.6 \pm 0.002$ | $0.737 \pm 0.009$ | $0.392 \pm 0.011$ | $0.619 \pm 0.015$ |
| | CP-ALS | $0.64 \pm 0.009$ | $0.745 \pm 0.008$ | $0.593 \pm 0.121$ | $0.637 \pm 0.015$ |
| Static | CT-CP | $0.459 \pm 0.005$ | $0.27 \pm 0.003$ | $0.488 \pm 0.005$ | $0.626 \pm 0.012$ |
| | CT-GP | $0.412 \pm 0.004$ | $0.282 \pm 0.23$ | $0.557 \pm 0.009$ | $0.628 \pm 0.01$ |
| | BCTT | $0.342 \pm 0.005$ | $0.157 \pm 0.015$ | $0.234 \pm 0.005$ | $0.581 \pm 0.022$ |
| | NONFAT | $0.335 \pm 0.002$ | $\mathbf{0.077 \pm 0.002}$ | $0.256 \pm 0.003$ | $0.627 \pm 0.005$ |
| | THIS-ODE | $0.362 \pm 0.002$ | $0.089 \pm 0.002$ | $0.357 \pm 0.007$ | $0.603 \pm 0.013$ |
| | POST | $0.461 \pm 0.008$ | $0.518 \pm 0.087$ | $0.357 \pm 0.011$ | $0.558 \pm 0.058$ |
| | ADF-CP | $0.451 \pm 0.006$ | $0.489 \pm 0.009$ | $0.384 \pm 0.014$ | $0.411 \pm 0.025$ |
| Stream | BASS | $0.745 \pm 0.026$ | $0.749 \pm 0.01$ | $0.903 \pm 0.044$ | $0.721 \pm 0.038$ |
| | SFTL-CP | $\mathbf{0.243 \pm 0.003}$ | $0.111 \pm 0.008$ | $0.159 \pm 0.004$ | $0.323 \pm 0.003$ |
| | SFTL-Tucker | $0.253 \pm 0.004$ | $0.273 \pm 0.033$ | $\mathbf{0.144 \pm 0.008}$ | $\mathbf{0.273 \pm 0.016}$ |

Table 6: Final prediction error with $R = 7$. The results were averaged from five runs.

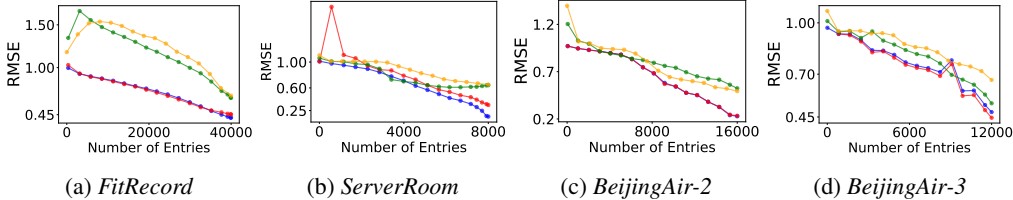

(a) *FitRecord*    (b) *ServerRoom*    (c) *BeijingAir-2*    (d) *BeijingAir-3*

Figure 7: Online prediction error with the number of processed entries ($R = 7$)

incremental updates and never needs to revisit the past data. To demonstrate this point, we compared the training time of our method with BCTT on *BeijingAir2* dataset. All the methods were run on a Linux workstation. From Table 7, we can see a large speed-up of our method with both the CP and Tucker form. The higher the rank ($R$), the more significant the speed-up.

| | $R = 2$ | $R = 3$ | $R = 5$ | $R = 7$ |
|---|---|---|---|---|
| SFTL-CP | 27.1 | 27.2 | 28.5 | 29.1 |
| SFTL-Tucker | 32.3 | 35.6 | 43.2 | 59.3 |
| BCTT | 49.5 | 56.1 | 72.1 | 136.7 |

Table 7: Running time in seconds on *BeijingAir2* dataset.

