| | RMSE | *FitRecord* | *ServerRoom* | *BeijingAir-2* | *BeijingAir-3* |
|---|---|---|---|---|---|
| Static | PTucker | 0.606 ± 0.015 | 0.757 ± 0.36 | 0.509 ± 0.01 | 0.442 ± 0.142 |
| | Tucker-ALS | 0.914 ± 0.01 | 0.991 ± 0.016 | 0.586 ± 0.016 | 0.896 ± 0.032 |
| | CP-ALS | 0.926 ± 0.013 | 0.997 ± 0.016 | 0.647 ± 0.041 | 0.918 ± 0.031 |
| | CT-CP | 0.675 ± 0.009 | 0.412 ± 0.024 | 0.642 ± 0.007 | 0.832 ± 0.035 |
| | CT-GP | 0.611 ± 0.009 | 0.218 ± 0.021 | 0.723 ± 0.01 | 0.88 ± 0.026 |
| | BCTT | 0.604 ± 0.019 | 0.715 ± 0.352 | 0.504 ± 0.01 | 0.799 ± 0.027 |
| | NONFAT | 0.543 ± 0.002 | **0.132 ± 0.002** | 0.425 ± 0.002 | 0.878 ± 0.014 |
| | THIS-ODE | 0.544 ± 0.005 | 0.142 ± 0.004 | 0.553 ± 0.015 | 0.876 ± 0.027 |
| Stream | POST | 0.705 ± 0.013 | 0.767 ± 0.155 | 0.539 ± 0.01 | 0.695 ± 0.135 |
| | ADF-CP | 0.669 ± 0.033 | 0.764 ± 0.114 | 0.583 ± 0.07 | 0.54 ± 0.045 |
| | BASS-Tucker | 1 ± 0.016 | 1 ± 0.016 | 1.043 ± 0.05 | 0.982 ± 0.058 |
| | SFTL-CP | **0.437 ± 0.014** | 0.18 ± 0.019 | **0.323 ± 0.019** | 0.462 ± 0.009 |
| | SFTL-Tucker | 0.446 ± 0.024 | 0.276 ± 0.031 | 0.344 ± 0.031 | **0.417 ± 0.035** |
| | MAE | | | | |
| Static | PTucker | 0.416 ± 0.005 | 0.388 ± 0.152 | 0.336 ± 0.004 | 0.271 ± 0.053 |
| | Tucker-ALS | 0.676 ± 0.008 | 0.744 ± 0.01 | 0.408 ± 0.008 | 0.669 ± 0.02 |
| | CP-ALS | 0.686 ± 0.011 | 0.748 ± 0.009 | 0.454 ± 0.057 | 0.691 ± 0.016 |
| | CT-CP | 0.466 ± 0.005 | 0.295 ± 0.029 | 0.49 ± 0.006 | 0.642 ± 0.02 |
| | CT-GP | 0.424 ± 0.006 | 0.155 ± 0.012 | 0.517 ± 0.01 | 0.626 ± 0.01 |
| | BCTT | 0.419 ± 0.015 | 0.534 ± 0.263 | 0.343 ± 0.003 | 0.579 ± 0.018 |
| | NONFAT | 0.373 ± 0.001 | **0.083 ± 0.001** | 0.282 ± 0.002 | 0.622 ± 0.006 |
| | THIS-ODE | 0.377 ± 0.003 | 0.097 ± 0.003 | 0.355 ± 0.008 | 0.606 ± 0.015 |
| Stream | POST | 0.485 ± 0.008 | 0.564 ± 0.091 | 0.368 ± 0.008 | 0.517 ± 0.123 |
| | ADF-CP | 0.462 ± 0.022 | 0.574 ± 0.073 | 0.401 ± 0.029 | 0.415 ± 0.038 |
| | BASS | 0.777 ± 0.039 | 0.749 ± 0.01 | 0.871 ± 0.125 | 0.727 ± 0.029 |
| | SFTL-CP | **0.248 ± 0.005** | 0.126 ± 0.007 | **0.199 ± 0.005** | 0.311 ± 0.004 |
| | SFTL-Tucker | 0.25 ± 0.01 | 0.203 ± 0.032 | 0.218 ± 0.02 | **0.261 ± 0.023** |

Table 3: Final prediction error with $R = 2$. The results were averaged from five runs.

| | RMSE | *FitRecord* | *ServerRoom* | *BeijingAir-2* | *BeijingAir-3* |
|---|---|---|---|---|---|
| Static | PTucker | 0.603 ± 0.045 | 0.677 ± 0.129 | 0.464 ± 0.012 | 0.421 ± 0.074 |
| | Tucker-ALS | 0.885 ± 0.007 | 0.989 ± 0.014 | 0.559 ± 0.017 | 0.863 ± 0.032 |
| | CP-ALS | 0.907 ± 0.015 | 0.993 ± 0.016 | 0.594 ± 0.031 | 0.901 ± 0.03 |
| | CT-CP | 0.666 ± 0.008 | 0.5 ± 0.2 | 0.641 ± 0.006 | 0.819 ± 0.019 |
| | CT-GP | 0.606 ± 0.008 | 0.217 ± 0.025 | 0.749 ± 0.014 | 0.895 ± 0.054 |
| | BCTT | 0.576 ± 0.015 | 0.358 ± 0.082 | 0.454 ± 0.011 | 0.829 ± 0.028 |
| | NONFAT | 0.517 ± 0.002 | **0.129 ± 0.002** | 0.408 ± 0.005 | 0.877 ± 0.014 |
| | THIS-ODE | 0.528 ± 0.005 | 0.132 ± 0.002 | 0.544 ± 0.014 | 0.878 ± 0.026 |
| Stream | POST | 0.706 ± 0.034 | 0.741 ± 0.161 | 0.518 ± 0.016 | 0.622 ± 0.123 |
| | ADF-CP | 0.641 ± 0.009 | 0.652 ± 0.012 | 0.542 ± 0.012 | 0.518 ± 0.003 |
| | BASS-Tucker | 1.008 ± 0.017 | 1 ± 0.016 | 1.035 ± 0.038 | 0.99 ± 0.034 |
| | SFTL-CP | 0.434 ± 0.014 | 0.178 ± 0.006 | **0.288 ± 0.017** | 0.454 ± 0.011 |
| | SFTL-Tucker | **0.418 ± 0.01** | 0.289 ± 0.096 | 0.314 ± 0.049 | **0.41 ± 0.013** |
| | MAE | | | | |
| Static | PTucker | 0.392 ± 0.009 | 0.323 ± 0.053 | 0.307 ± 0.005 | 0.197 ± 0.029 |
| | Tucker-ALS | 0.648 ± 0.012 | 0.743 ± 0.008 | 0.39 ± 0.008 | 0.651 ± 0.018 |
| | CP-ALS | 0.666 ± 0.013 | 0.746 ± 0.01 | 0.415 ± 0.022 | 0.676 ± 0.021 |
| | CT-CP | 0.462 ± 0.005 | 0.348 ± 0.141 | 0.489 ± 0.006 | 0.632 ± 0.015 |
| | CT-GP | 0.419 ± 0.005 | 0.158 ± 0.022 | 0.544 ± 0.012 | 0.627 ± 0.015 |
| | BCTT | 0.392 ± 0.004 | 0.267 ± 0.067 | 0.299 ± 0.006 | 0.607 ± 0.027 |
| | NONFAT | 0.355 ± 0.001 | **0.078 ± 0.001** | 0.265 ± 0.003 | 0.622 ± 0.006 |
| | THIS-ODE | 0.363 ± 0.004 | 0.083 ± 0.002 | 0.348 ± 0.006 | 0.603 ± 0.009 |
| Stream | POST | 0.482 ± 0.022 | 0.54 ± 0.102 | 0.351 ± 0.009 | 0.442 ± 0.109 |
| | ADF-CP | 0.445 ± 0.006 | 0.5 ± 0.009 | 0.381 ± 0.006 | 0.393 ± 0.009 |
| | BASS | 0.822 ± 0.024 | 0.749 ± 0.009 | 0.919 ± 0.041 | 0.73 ± 0.018 |
| | SFTL-CP | 0.246 ± 0.005 | 0.121 ± 0.003 | **0.176 ± 0.006** | 0.305 ± 0.006 |
| | SFTL-Tucker | **0.24 ± 0.002** | 0.18 ± 0.042 | 0.196 ± 0.03 | **0.263 ± 0.011** |

Table 4: Final prediction error with $R = 3$. The results were averaged from five runs.

| | RMSE | *FitRecord* | *ServerRoom* | *BeijingAir-2* | *BeijingAir-3* |
|---|---|---|---|---|---|
| Static | PTucker | $0.603 \pm 0.045$ | $0.677 \pm 0.129$ | $0.464 \pm 0.012$ | $0.421 \pm 0.074$ |
| | Tucker-ALS | $0.826 \pm 0.003$ | $0.983 \pm 0.016$ | $0.586 \pm 0.018$ | $0.825 \pm 0.026$ |
| | CP-ALS | $0.878 \pm 0.012$ | $0.994 \pm 0.013$ | $0.897 \pm 0.215$ | $0.863 \pm 0.024$ |
| | CT-CP | $0.663 \pm 0.008$ | $0.384 \pm 0.008$ | $0.64 \pm 0.007$ | $0.818 \pm 0.019$ |
| | CT-GP | $0.603 \pm 0.006$ | $0.381 \pm 0.303$ | $0.766 \pm 0.016$ | $0.904 \pm 0.046$ |
| | BCTT | $0.498 \pm 0.011$ | $0.194 \pm 0.017$ | $0.368 \pm 0.01$ | $0.813 \pm 0.028$ |
| | NONFAT | $0.497 \pm 0.003$ | $\mathbf{0.128 \pm 0.002}$ | $0.394 \pm 0.004$ | $0.88 \pm 0.013$ |
| | THIS-ODE | $0.138 \pm 0.003$ | $0.554 \pm 0.016$ | $0.878 \pm 0.027$ | |
| Stream | POST | $0.675 \pm 0.012$ | $0.707 \pm 0.14$ | $0.519 \pm 0.017$ | $0.738 \pm 0.068$ |
| | ADF-CP | $0.652 \pm 0.01$ | $0.646 \pm 0.008$ | $0.548 \pm 0.012$ | $0.552 \pm 0.026$ |
| | BASS-Tucker | $0.604 \pm 0.043$ | $0.493 \pm 0.071$ | $0.391 \pm 0.005$ | $0.634 \pm 0.083$ |
| | SFTL-CP | $\mathbf{0.424 \pm 0.006}$ | $0.166 \pm 0.013$ | $0.256 \pm 0.013$ | $0.481 \pm 0.006$ |
| | SFTL-Tucker | $0.448 \pm 0.009$ | $0.406 \pm 0.052$ | $\mathbf{0.249 \pm 0.017}$ | $\mathbf{0.432 \pm 0.019}$ |
| | MAE | | | | |
| Static | PTucker | $0.353 \pm 0.005$ | $0.305 \pm 0.042$ | $0.248 \pm 0.004$ | $0.32 \pm 0.038$ |
| | Tucker-ALS | $0.6 \pm 0.002$ | $0.737 \pm 0.009$ | $0.392 \pm 0.011$ | $0.619 \pm 0.015$ |
| | CP-ALS | $0.64 \pm 0.009$ | $0.745 \pm 0.008$ | $0.593 \pm 0.121$ | $0.637 \pm 0.015$ |
| | CT-CP | $0.459 \pm 0.005$ | $0.27 \pm 0.003$ | $0.488 \pm 0.005$ | $0.626 \pm 0.012$ |
| | CT-GP | $0.412 \pm 0.004$ | $0.282 \pm 0.23$ | $0.557 \pm 0.009$ | $0.628 \pm 0.01$ |
| | BCTT | $0.342 \pm 0.005$ | $0.157 \pm 0.015$ | $0.234 \pm 0.005$ | $0.581 \pm 0.022$ |
| | NONFAT | $0.335 \pm 0.002$ | $\mathbf{0.077 \pm 0.002}$ | $0.256 \pm 0.003$ | $0.627 \pm 0.005$ |
| | THIS-ODE | $0.362 \pm 0.002$ | $0.089 \pm 0.002$ | $0.357 \pm 0.007$ | $0.603 \pm 0.013$ |
| Stream | POST | $0.461 \pm 0.008$ | $0.518 \pm 0.087$ | $0.357 \pm 0.011$ | $0.558 \pm 0.058$ |
| | ADF-CP | $0.451 \pm 0.006$ | $0.489 \pm 0.009$ | $0.384 \pm 0.014$ | $0.411 \pm 0.025$ |
| | BASS | $0.745 \pm 0.026$ | $0.749 \pm 0.01$ | $0.903 \pm 0.044$ | $0.721 \pm 0.038$ |
| | SFTL-CP | $\mathbf{0.243 \pm 0.003}$ | $0.111 \pm 0.008$ | $0.159 \pm 0.004$ | $0.323 \pm 0.003$ |
| | SFTL-Tucker | $0.253 \pm 0.004$ | $0.273 \pm 0.033$ | $\mathbf{0.144 \pm 0.008}$ | $\mathbf{0.273 \pm 0.016}$ |

Table 5: Final prediction error with $R = 7$. The results were averaged from five runs.

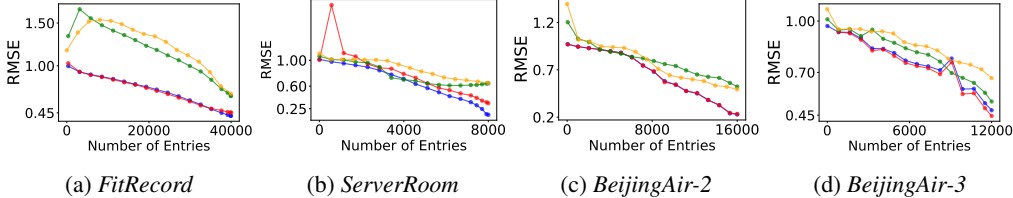

(a) *FitRecord*  (b) *ServerRoom*  (c) *BeijingAir-2*  (d) *BeijingAir-3*

Figure 3: Online prediction error with the number of processed entries ($R = 7$)

the training time of our method with BCTT on *BeijingAir2* dataset. All the methods ran on a Linux workstation. From Table 6, we can see a large speed-up of our method with both the CP and Tucker form. The higher the rank ($R$), the more significant the speed-up.

| | $R = 2$ | $R = 3$ | $R = 5$ | $R = 7$ |
|---|---|---|---|---|
| SFTL-CP | 27.1 | 27.2 | 28.5 | 29.1 |
| SFTL-Tucker | 32.3 | 35.6 | 43.2 | 59.3 |
| BCTT | 49.5 | 56.1 | 72.1 | 136.7 |

Table 6: Running time in seconds on *BeijingAir2* dataset.

# H  Limitation and Discussion

The state-space prior used our method arises from the LTI-SDE (7), an equivalent representation of the GP prior over time functions using a type of Matérn kernels. While elegant and useful, building equivalent SDEs to a specific GP prior might restrict the expressivity of our model. To overcome this limitation, we plan to construct an SDE prior directly, *e.g.,* a linear SDE to model how the factor

trajectory varies along the time. Then we consider converting the SDE into a state-space prior. In doing so, we can further improve the flexibility of our model to capture more complex temporal evolution, *e.g.,* non-stationary and highly fluctuating.