# OpenReview forum: "Streaming Factor Trajectory Learning for Temporal Tensor Decomposition"
_NeurIPS.cc/2023/Conference — NeurIPS 2023 poster_

### Official Review · Reviewer_Dugj · 2023-06-13

**Soundness:** 3 good
**Presentation:** 4 excellent
**Contribution:** 3 good
**Rating:** 6
**Confidence:** 3

**Summary:**

In traditional tensor decomposition, the factor of decomposition is static. Therefore, it was not possible to fit time-evolving data well. This paper develops SFTL using a Gaussian process to capture the time-developing trajectory of the factors. Their naive implementation would have increased computational complexity. Therefore, they reduce the computational complexity by replacing the Matern kernel in GP with an SDE.

**Strengths:**

- Standard tensor decomposition can only extract linear patterns. SFTL, however, uses a Gaussian process to extract nonlinear features in the time direction.
- Elegant techniques are used to reduce the computational complexity. It is unfortunate that these techniques are not applicable to non-matern kernels, i.e., RBF kernels.
- They provide a theoretical discussion of the computational complexity of the proposed method and include additional experimental results in the appendix.
- Notation of the symbols is clear and not misleading. The paper is well written.
- It is good to have a discussion about the sensitivity of hyperparameters.

**Weaknesses:**

There are some known tricks to reduce the computational complexity of the Gaussian process from O(n^3), such as Inducing Variable Method, Deterministic Training Conditional (DTC) Approximation, and Fully Independent Training Conditional (FITC) Approximation. All of these tricks can be adapted to non-Matern kernels. With these tricks, I think it is possible to achieve fast computations without Hartikainen's elegant tricks. Although the elegant method of this paper is to be evaluated, it is a drawback that there is no comment on this point.

**Questions:**

- I struggled to understand Fig 1. Since there are 3 trajectories in Fig 1, does this correspond to the SFTL for a tensor that has three 3 modes?
- In my understanding, the size of vec(z) depends on the rank R, where R is a hyperparameter. In Algorithm 1, R is not included in Input. That point needs to be clarified.
- In section 6.1, the word `node` suddenly appears, but I am unclear what this means. My understanding is that the input to the experiment in section 6.1 is a full-rank 2x2 time-evolving matrix with noise, is this correct?

**Limitations:**

The authors appropriately mention in the Appendix that the speedup of the proposed method is only applicable to Matern kernels.

---

> ### Author Rebuttal · Authors · 2023-08-06
>
> Thanks for your insightful and constructive comments. Here are our responses. C: comments; R: responses
>
> >C1: "There are some known tricks to reduce the computational complexity of the Gaussian process from O(n^3), ... I think it is possible to achieve fast computations without Hartikainen's elegant tricks. Although the elegant method of this paper is to be evaluated, it is a drawback that there is no comment on this point.
>
> R1: Great comments!  We do agree these methods, like DTC and FITC, can reduce the GP complexity as well. However, these methods have to use additional approximations, namely, a low-rank or sparse approximation of the kernel matrix to reduce the cost, which might bring additional errors and learning challenges.  Our motivation of using LTI-SDE is twofold: (1) we can completely avoid sparse/low-rank approximations and perform exact GP inference with linear time complexity. (2) With the state-space prior, we can fulfill efficient streaming inference with the online filtering and smoothing framework. On the contrary, conducting streaming inference for sparse latent GP models is much more challenging (e.g., how to dynamically update the inducing points and locations). Thanks for the constructive suggestion. We will add the discussion into our paper.
>
> >C2: "I struggled to understand Fig 1. Since there are 3 trajectories in Fig 1, does this correspond to the SFTL for a tensor that has three 3 modes?"
>
> R2: Yes, Fig. 1 indeed uses a tensor with 3 modes as an example to illustrate the conditional independence in our model. Please see global response R1 for our clarification of Fig. 1. We will add more detailed explanation of Fig.1 in our paper.
>
> >C3: "In Algorithm 1, R is not included in Input. That point needs to be clarified."
>
> R3: Thanks for the great comment and suggestion! We do agree. We will add $R$ into the input of our algorithm table.
>
> >C4: "In section 6.1, the word node suddenly appears, but I am unclear what this means. My understanding is that the input to the experiment in section 6.1 is a full-rank 2x2 time-evolving matrix with noise, is this correct?"
>
> R4: We appreciate the reviewer catching this problem. Yes, your understanding is correct. "node" means the object in the mode, and we should use the word "object" to avoid any confusion. We will modify them to improve the clarify.

---

> > ### Comment · Reviewer_Dugj · 2023-08-11
> >
> > Thank you for your response. There is one point I would like to clarify.
> >
> > > perform exact GP inference with linear time complexity
> >
> > Does this mean that the time complexity is linear with respect to the time size $L$? Or does it mean that the time complexity is linear for some other quantity? I am not an expert in computational complexity theory so I am confused by the wording and would appreciate it if you could explain this to me.

---

> > > ### Author Response · Authors · 2023-08-11
> > >
> > > Thanks for the question. Yes, exactly, the linear time complexity means it is linear with respect to the number of time points L --- since our the trajectory is a function of time, and these time points form the inputs to the GP model. When we convert the GP to the LTI-SDE and then the state-space prior, each state is associated with the a time point. The computation in the inference, including both filtering and RTS smoother, is linear with the number of states. So overall, it is linear time complexity with L.

---

> > > > ### Comment · Reviewer_Dugj · 2023-08-11
> > > >
> > > > I appreciate your responses. Now, it made sense.
> > > >
> > > > Your comments are very reasonable. As you said, adding the discussion related to C1 in the camera-ready version will strengthen this paper.
> > > >
> > > > While it would be more impressive if you could evaluate how much accuracy improves by not using these approximations (DTC and FITC etc.), I am satisfied with the manuscript, so I will keep the score.

---

> > > > > ### Author Response · Authors · 2023-08-11
> > > > >
> > > > > Thank you! We will follow your suggestions to improve our paper.

---

### Official Review · Reviewer_ZhxY · 2023-07-05

**Soundness:** 2 fair
**Presentation:** 2 fair
**Contribution:** 2 fair
**Rating:** 6
**Confidence:** 2

**Summary:**

The authors highlight that real-world tensor data often includes time information, representing the timestamps of interactions between objects in different modes. However, most existing decomposition methods for tensors with temporal information only estimate static factors for each object and cannot capture the temporal variation of the factors. To address this limitation, the authors introduce Streaming Factor Trajectory Learning (SFTL), which can efficiently handle data streams and estimate the temporal evolution of objects' representation without storing or re-accessing previous data.

They use a Gaussian process (GP) prior to model the factor trajectory of each object as a function of time. GPs are flexible nonparametric function priors that can capture complex temporal dynamics. The factor trajectories are combined using either the CANDECOMP/PARAFAC (CP) or Tucker decomposition to sample the tensor entry values at any given time.

The authors evaluate SFTL using both simulation studies and real-world temporal tensor datasets. In the simulation studies, SFTL recovers nonlinear factor trajectories and provides reasonable uncertainty estimation. When applied to real-world datasets for missing value prediction, SFTL consistently outperforms state-of-the-art streaming CP and Tucker decomposition algorithms in both online and final predictive performance. In some cases, SFTL even surpasses the prediction accuracy of recent static decomposition methods that require multiple passes through the dataset. The learned factor trajectories from a real-world dataset also exhibit interesting temporal evolution.

**Strengths:**

The main advantage of this method is its computational efficiency. To overcome the computational challenges of the GP's covariance matrix computation in streaming inference, they employ spectral analysis to convert the GP into a linear time-invariant (LTI) stochastic differential equation (SDE). This conversion enables them to transform the SDE into an equivalent state-space prior over the factor states at observed timestamps, simplifying posterior inference and making it computationally efficient.

In addition, they increase the scalability of tensor decomposition as in the following. They leverage the chain structure of the state-space prior and utilize the conditional expectation propagation framework to develop an efficient online filtering algorithm. This algorithm allows for the estimation of a decoupled running posterior of the involved factor states when new entries at a new timestamp arrive. The decoupled Gaussian estimate enables independent and parallel computation of the full posterior of each factor trajectory without revisiting previous data. The method scales linearly with the number of observed timestamps.


**Weaknesses:**

I think the paper is not so clear and it is hard to follow the contribution especially in Section 3 and 4. The simulation study looks fine, but not a big difference is observed between SOTA and the STL on the real-world applications.  Besides the real-world applications are not explained, so it is also not clear to understand what are the application areas of this method.

**Questions:**

Can you please explain the real-world applications (we only see the dataset names, but it is not sufficient to understand the application)?
Can you also add how much gain you provide with STL compared to baselines, both in efficiency and accuracy, e.g. in percent?
Can you please add more explanations to Figure 1? It is not so easy to understand how the overall method works from that figure.


**Limitations:**

The authors can discuss more the scalability and the applicability of STL. What are the applications that STL can be used in real-world?

---

> ### Author Rebuttal · Authors · 2023-08-06
>
> Thanks for your comments and questions. Here are our clarification and responses. Please see global response R1 for our clarification of Fig. 1.  C: comments; R: responses
>
> > C1: "the real-world applications are not explained, so it is also not clear to understand what are the application areas of this method. What are the applications that STL can be used in real-world? Can you please explain the real-world applications?"
>
> R1: Thanks for the question. In fact, we did explain the application and dataset details  in **Section E of Appendix**. Due to the space limit, we put this information in Appendix, but we point out the place in Line 284 of the main paper. *FitRecord* is about the outdoor exercises, *ServerRoom* is about the temperature management for a supercomputing center, and *BeijingAir-2* and *BeijingAir-3* are about the air pollutant monitoring. Please see the details in Section E of Appendix.
>
> > C2: The simulation study looks fine, but not a big difference is observed between SOTA and the STL on the real-world applications...Can you also add how much gain you provide with STL compared to baselines, both in efficiency and accuracy, e.g. in percent?
>
> R2: In fact, we do view the difference between SOTA and SFTL is **big**.  First, as a streaming factorization approach, SFTL consistently outperforms its counterparts --- **the state-of-the-art streaming tensor decomposition algorithms** --- by a large margin in all the ranks. Please see Table 1 of the main paper, and Table 3-5 of Appendix. To highlight the difference, we follow your suggestion to list the performance gain of our method over the best competing approaches in RMSE as follows. As we can see, for every rank, SFTL achieves at least 16% improvement. In most cases, the improvement is over 30%.
>
> |  FitRecord   | R=2 |   R=3 |     R=5 |    R=7 |
> | :---        |            ---: |---:  | ---:  | ---:  |
> | Best (streaming)    | 0.669   | 0.641   |  0.648  | 0.604   |
> | SFTL  | 0.437      | 0.418  | 0.424   | 0.424    |
> | Performance Gain  |   36%    | 35%   | 37%   | 30%   |
>
> |  ServerRoom   | R=2 |   R=3 |     R=5 |    R=7 |
> | :---        |            ---: |---:  | ---:  | ---:  |
> | Best (streaming)    | 0.764   |  0.652   |  0.64  | 0.593  |
> | SFTL  | 0.18      | 0.178   | 0.161   | 0.166   |
> | Performance Gain   |   76%    | 72%   | 75%   | 72%   |
>
> |  BeijingAir2   | R=2 |   R=3 |     R=5 |    R=7 |
> | :---        |            ---: |---:  | ---:  | ---:  |
> | Best (streaming)    | 0.539   |  0.518   |   0.516  |  0.491  |
> | SFTL  | 0.323     | 0.288   | 0.248   | 0.249   |
> | Performance Gain   |   41%    | 43%   | 52%   | 48%   |
>
> |  BeijingAir3   | R=2 |   R=3 |     R=5 |    R=7 |
> | :---        |            ---: |---:  | ---:  | ---:  |
> | Best (streaming)    |  0.54   |  0.622   |   0.551  |  0.552  |
> | SFTL  | 0.417     | 0.41  | 0.439   |   0.432  |
> | Performance Gain   |   24%    | 34%   | 22%   | 16%   |
>
> Second, even compared with non-streaming (i.e., static) decomposition methods, in most cases, SFTL still exhibits considerable improvement, except that on ServerRoom dataset, SFTL is slightly worse than NONFAT and THIS-ODE yet is still much better than all the other static decomposition methods. Note that `the static methods can repeatedly access the dataset for many training epochs, while the streaming method can only pass through the dataset for once`. Hence, the static methods have `much more privilege on accessing training information`. It is therefore not surprising to expect that the static methods should perform better in prediction accuracy than streaming methods. However, **SFTL can still outperforms the static methods in most cases and often by a large margin, which further shows the advantage of SFTL as a streaming approach**.  To highlight this point, we list the performance gain over the static methods in RMSE as follows.
>
> |  RMSE (R=2)  | FitRecord |   BeijingAir-2 |     BeijingAir-3 |
> | :---        |            ---: |---:  | ---:  |
> | Best (static methods)    | 0.543   | 0.425   |   0.442  |
> | SFTL  | 0.437      | 0.323   | 0.417   |
> | Performance Gain   |   20%    | 24%   | 8%   |
>
> |  RMSE (R=3)  | FitRecord |   BeijingAir-2 |     BeijingAir-3 |
> | :---        |            ---: |---:  | ---:  |
> | Best (static methods)    | 0.517  | 0.408   |  0.421  |
> | SFTL  | 0.418      | 0.288   | 0.41   |
> | Performance Gain   |   19%    | 30%   | 3%   |
>
> |  RMSE (R=5)  | FitRecord |   BeijingAir-2 |     BeijingAir-3 |
> | :---        |            ---: |---:  | ---:  |
> | Best (static methods)    | 0.503   | 0.395   |  0.535  |
> | SFTL  | 0.424      | 0.248   | 0.439   |
> | Performance Gain   |   16%    | 38%   | 19%   |
>
> |  RMSE (R=7)  | FitRecord |   BeijingAir-2 |     BeijingAir-3 |
> | :---        |            ---: |---:  | ---:  |
> | Best (static methods)    | 0.497   | 0.394   |  0.451  |
> | SFTL  | 0.424      | 0.249   | 0.432   |
> | Performance Gain   |   16%    | 37%   | 6%   |
>
> To examine the efficiency gain, we extend our running time results in Table 7 of Appendix by adding the speed-up percentage. We compared with the running time of BCTT on BeijingAir2 dataset; please see Section G of Appendix for details.
>
> |  Training Time(Sec)   | R=2 |   R=3 |     R=5 |    R=7 |
> | :---        |            ---: |---:  | ---:  | ---:  |
> | BCTT (benchmark)    | 49.5   | 56.5   |  72.1  | 136.7  |
> | SFTL-Tucker  | 32.3      | 35.6   | 43.2   | 59.3   |
> | SFTL-CP  | 27.1      | 27.2   | 28.5   | 29.1 |
> | Speed-up SFTL-Tucker   |   55%    | 59%   | 67%   | 131%   |
> | Speed-up SFTL-CP   |   83%    | 107%   | 152%   | 371%   |
>
> As we can see, the speed-up is significant as well. The larger the rank, and the bigger the speed-up. We will add these tables into our paper.

---

> ### Author Response · Authors · 2023-08-15
>
> Dear Reviewer:
>
> Could we know if the responses above have addressed your concerns/questions and if there is additional feedback that need us to respond? Thanks for your time.

---

> > ### Comment · Reviewer_ZhxY · 2023-08-16
> > **Response to the rebuttal**
> >
> > Thank you for bringing these points to my attention and addressing the comments. After reading the responses and other reviews carefully, I decided to increase my score.

---

> > > ### Author Response · Authors · 2023-08-16
> > >
> > > Thanks for your feedback!

---

### Official Review · Reviewer_vq55 · 2023-07-06

**Soundness:** 3 good
**Presentation:** 2 fair
**Contribution:** 3 good
**Rating:** 6
**Confidence:** 3

**Summary:**

This paper studies the dynamic tensor factorization under the streaming data setting. In particular, the authors propose to use a state-space Gaussian process (GP) to parameterize the CP or Tucker factors as functions of time. To efficiently learn the GP and handle streaming data, an equivalent SDE and a conditional expectation propagation algorithm are then derived. For experiments, the authors show that the model can learn factor and tensor entry trajectories. Besides, it achieves good competition performance, even compared with model non-streaming settings.

**Strengths:**

1. This work presents the first model that attempts to learn factor trajectories under streaming data setting.

2. The performance of completion is good. Even though the proposed model runs under the online setting, the error is smaller than many models that use the whole dataset.


**Weaknesses:**

1. Although the problem setting is new in the tensor decomposition field, most techniques used in this work seem standard, such as state-space Gaussian process and expectation propagation.

2. The notations are too complex and hard to follow. There are too many super/sub-scripts and sometimes the meanings are unclear. For examples, for latent variables $z$, sometimes, the second subscript is the rank (e.g., in Line 128, $z^m_{j, r}(t)$, where $r$ is the rank). Then, in Line 141, it becomes the time stamp, e.g., $\bar{z}^m_{j, k}$ with $k$ being the time index.


**Questions:**

1. In Equation 9, for the first Gaussian distribution, is the variance missing?

---

> ### Author Rebuttal · Authors · 2023-08-06
>
> Thanks for your valuable comments. Here are our responses. C: comments; R: response
>
> > C1: "Although the problem setting is new in the tensor decomposition field, most techniques used in this work seem standard, such as state-space Gaussian process and expectation propagation."
>
> R1: We do agree that we did not invent new techniques from scratch. However, many works, including the many references in Related Work, also use existent techniques, via their novel application and combination, to address the problem of interest. we believe such works are still valuable contributions to the relevant fields. In addition to the new problem setting, we believe the novelty of our work is mainly threefold: (1) We are the first to apply the LTI-SDE representation of GPs to learn the factor trajectories for tensor decomposition. Not only is this modeling a nice fit to our problem, it also evades the computational bottleneck and various sparse approximations, which are the known headaches for GP learning. (2) We combined the state-space model and CEP framework to develop an efficient filtering algorithm, which overcomes the infeasibility of the standard Kalman filtering and allows fast fusion of the incremental data information. (3) Our novel approximation structure (Eq6) further enables the application of the classical RTS smoother, which can update the trajectory posterior highly efficiently and in parallel (this is critical to fast streaming inference). We have demonstrated the effectiveness of our method in a series of benchmark datasets.
>
> > C2: "The notations are too complex and hard to follow."
>
> R2: Thanks for the comments and we agree. To introduce the state and trajectory values of each factor, we need to index the tensor mode, the object, the rank and the time, which indeed make the notation cluttered. We will introduce more concise notations, and avoid using similar symbols, like $z$ and $\bar{z}$, to denote different variables. We will further polish our paper to make it more friendly to readers.
>
> > C3: "In Equation 9, for the first Gaussian distribution, is the variance missing?"
>
> R3: Yes, and thanks much for catching this typo! We will fix it in the paper.

---

> ### Author Response · Authors · 2023-08-16
>
> Dear Reviewer vq55,
>
> Could we kindly know if the responses have addressed your concerns and if further explanations or clarifications are needed? Your time and efforts in evaluating our work are appreciated greatly.

---

> > ### Comment · Reviewer_vq55 · 2023-08-16
> >
> > Thanks for your responses.
> > Considering the great improvements in empirical evaluations,
> > I think this method could contribute to the related community.
> > Especially, the proposed model outperforms many static decomposition baselines,
> > which are under much easier settings in general.
> > I would like to raise my score to 6.
> > However, I think simplifying the notations is still essential to make this paper more clear.

---

> > > ### Author Response · Authors · 2023-08-16
> > >
> > > We appreciate your feedback. We will surely simplify the notation to improve the clarity of our paper.

---

### Official Review · Reviewer_DKpV · 2023-07-07

**Soundness:** 3 good
**Presentation:** 4 excellent
**Contribution:** 2 fair
**Rating:** 6
**Confidence:** 3

**Summary:**

The work proposes a new approach for the decomposition of a tensor with a temporal dimension. The developed method can apply to streaming data where decomposing the entire tensor from scratch would be prohibitive.

The key idea is to model each entry (or parameter) in the decomposition factor as a function of time, thus representing a factor trajectory in time, and to assume that this function is sampled from a Gaussian process. Then, given the factor trajectories, initial tensor values are recovered by sampling from a specific Normal distribution.

**Strengths:**

The paper is well-written and easy to follow. The authors clearly state the necessary background, problem, and notations.
The considered problem of handling streaming data is relevant. However, many technical choices were left unjustified.

**Weaknesses:**

The primary concerns are about (implicit) assumptions made in the paper:
- Parameters of the decomposition factor are sampled from a GP with the same kernel. That says that the smoothness of the factor trajectories is assumed to be similar. That might be a restrictive assumption on the data the method is applied for, but it is not discussed.
- Relates to the previous concern. Continuity in the latent space of the factors translates to the continuity in the space of the initial tensor values that follow from (1). In practice, a slight change in the factors might hugely affect the initial tensor values and vice versa. It would be essential to see the results of the method on more diverse problems, e.g., streaming point cloud data.
- Finally, the trade-off between the decomposition rank and sustaining several Gaussian processes (i.e., better modeling of the factors). The rank R used in the experiments is relatively small.


Some of the technical choices for the method are not well justified:
- why is Gamma distribution used?
- how are the hyperparameters of the models chosen? Do the data dictate those, or is there a reason for them?

A simple baseline would tackle the problem as a regression task, thus possibly learned by a neural network.  It would be great to add it as a baseline.

**Questions:**

- The authors use CP and Tucker decomposition. Why Tensor Train or Tensor Ring decompositions, are not compared to or not used?
For example, [1] propose a simple method incorporating temporal component into the decomposition. It is interesting to see the number of parameters comparison, compression time, and quality metrics.

[1] T4DT: Tensorizing Time for Learning Temporal 3D Visual Data
Mikhail Usvyatsov, Rafael Ballester-Rippoll, Lina Bashaeva, Konrad Schindler, Gonzalo Ferrer, Ivan Oseledets

**Limitations:**

The limitations arise from the set of assumptions made by the methods. Is this method applicable to the streaming 3D visual data, where the change in scenes might be significant from scene to scene? How the chosen rank R affects the quality, and what is the trade-off for it? These questions are not discussed in the paper though critical for the method to be applicable in practice.

---

> ### Author Rebuttal · Authors · 2023-08-06
>
> Thanks for your detailed and insightlful comments! Here are our responses. Please see global responses R2 & R3 for your questions regarding regression baselines and Tensor Train and Tensor Ring decompositions. C: comments; R: responses.
>
> >C1: "Parameters of the decomposition factor are sampled from a GP with the same kernel. That says that the smoothness of the factor trajectories is assumed to be similar. That might be a restrictive assumption on the data the method is applied for, but it is not discussed.”
>
> R1: Great comment and we do agree. For the convenience of development and experiments, we used the same kernel for all the GP priors. This can indeed restrict the family of trajectory functions that can be learned. Actually, our model is flexible enough to incorporate different kernels, and it is straightforward to modify our algorithm to support multiple kernels (namely, constructing a different LTI-SDE for GPs with a different kernel and then running the same inference procedure).
>
> On the other hand, since we used the Matern kernel, which has already covered a large family, and is way more flexible than other popular kernels, like Gaussian kernel. For example, the Matern kernel with $\nu=3/2$ covers all the functions that are one-time differentiable, while the Gaussian kernel only models functions that are infinitely differentiable. From our experiments, a single Matern kernel has already been able to show the advantage of our method. But we do agree that introducing multiple kernels can potentially improve the performance further.
>
> Thanks for the great suggestion and we will supplement this discussion into our paper.
>
> >C2: "Continuity in the latent space of the factors translates to the continuity in the space of the initial tensor values that follow from (1). In practice, a slight change in the factors might hugely affect the initial tensor values and vice versa ...  Is this method applicable to the streaming 3D visual data, where the change in scenes might be significant from scene to scene?
>
> R2: We appreciate the reviewer raising this interesting point. We believe SFTL is able to handle the case reasonably well.
>
> Specifically, whenever the observation space exhibits a significant pattern change, SFTL will update the factor trajectories via the proposed online filtering and RTS smoother. The online filtering (via moment matching and CEP) integrates the information of the previous states and the newly observed data to construct a new state distribution (i.e, the running posterior). That is, the new state distribution is *not* created standalone to absorb the abrupt data change; it is melt with the previous state information. The RTS smoother further back-propagates the new state information to all the old states, and thereby *smooths out* the estimate of the entire trajectory. One can see the online filtering and RTS smoother fuse the observation information (which might contain a drastic pattern change) in a soft and global way, and the new information is spread into the entire state chain. In this way, the learning of the factor trajectories is robust and smooth.
>
> We thank the reviewer for the great question and will investigate in-depth this problem in our future work.
>
> >C3: "The rank R used in the experiments is relatively small. How the chosen rank R affects the quality, and what is the trade-off for it?"
>
> R3: We appreciate the reviewer's comment. The relatively small rank R used in the experiments is due to the intrinsic low-rank property of our datasets (see details in Section E of Appendix) - either being sparse (e.g. FitRecord) or with a small number of objects in each mode (ServerRoom, BeijingAir). Hence, we believe the lower rank setting is sufficient to capture their temporal patterns.
>
> We do appreciate with the reviewer's concern about "the trade-off between the decomposition rank and sustaining several Gaussian processes". In fact, we have examined SFTL with different ranks and compared the predictive performance and efficiency ($R \in \{2, 3, 5, 7\}$) to show the trade-off; see Table 1 in the main paper and Table 3-6 in Appendix for the detailed results and discussions.  We will test SFTL with denser data, such as streaming point cloud data, and higher ranks to investigate this trade-off more comprehensively. We thank the reviewer for the constructive suggestion.
>
> >C4: "why is Gamma distribution used?"
>
> R4: In Bayesian modeling, the Gamma distribution is the mostly widely used prior distribution to model the noise (inverse variance) for Gaussian models/likelihoods. We followed this standard choice to build a full Bayesian model for factor trajectory learning. Thanks for the question. We will clarify this point in our paper.
>
> >C5: "how are the hyperparameters of the models chosen? Do the data dictate those, or is there a reason for them?"
>
> R5: We used cross-validation to tune and select the hyper-parameters and hence "the data dictate those" (see the details in Line 285-289). In addition, we investigated how sensitive the predictive performance is to the hyper-parameter choice (the amplitude $a$ and the length-scale $\rho$). We refer the reviewer to Section D.2 (Sensitive Analysis) and Table 2 of Appendix for the detailed results.

---

> > ### Comment · Area_Chair_Bvtp · 2023-08-17
> > **what did you think of the authors' response?**
> >
> > The authors have provided a detailed response. Has it sufficiently addressed your concerns about their choice of priors and hyperparameters? If so, please revise your score and review to address it, and please acknowledge that you have read the authors' carefully written response regardless.

---

> > > ### Comment · Reviewer_DKpV · 2023-08-22
> > > **Follow-up on authors' response**
> > >
> > > I want to thank the authors for their responses.
> > >
> > > The authors resolved my questions and added the empirical evaluations I asked for.
> > > I would be interested to see the results of the dense data. The proposed approach might not work as nicely as with the low-rak data presented in the paper, and it is important to add such experiment results to see the application border of the method. Please add it to the final version of the paper.
> > >
> > > Given this, I would like to raise my score to 6.

---

> ### Author Response · Authors · 2023-08-16
>
> Dear Reviewer DKpV,
>
> Could we kindly know if the responses have addressed your concerns and if further explanations or clarifications are needed? Your time and efforts in evaluating our work are appreciated greatly.

---

### Author Rebuttal · Authors · 2023-08-06

We thank all for the reviewers for their careful, valuable and constructive comments! We post our global response to a few questions here. C:comments; R: responses

>C1: more explanation/clarification for Figure 1.

R1: Figure 1 explains the key step in our development of the online filtering algorithm. Whenever a new batch of observations $\mathcal{B}\_{n+1} $ is received, we want to compute the running posterior of the related trajectory states $\mathcal{\Theta}\_{n+1}$ (those are the factor states for objects showing up in $\mathcal{B}\_{n+1}$ ), which we denote by $p(\mathcal{\Theta}\_{n+1} | \mathcal{D}\_{t\_n}\cup \mathcal{B}\_{n+1})$. Here $\mathcal{D}\_{t_n}$ is all the data received before $\mathcal{B}\_{n+1}$, i.e., the past data. Our key observation is that --- as shown in Fig. 1 --- **the new states $\mathcal{\Theta}\_{n+1}$ and new data $\mathcal{B}\_{n+1}$ are  independent to the past data $\mathcal{D}\_{t_n}$, conditioned on the most-recent states (one-step earlier) $\mathcal{\Theta}\_n$ and noise inverse variance $\tau$** (in the Gaussian likelihood; see Eq1). In other words, (roughly speaking) if we know the most recent states $\mathcal{\Theta}\_n$ and $\tau$, we do not need to access the past data $\mathcal{D}\_{t_n}$ to learn the new states $\mathcal{\Theta}\_{n+1}$. This is due to the chain structure between the successive states in the state-space prior; see Eq4. Fig. 1 illustrates this observation from the graphical model representation. We can see that **conditioned on $\mathcal{\Theta}\_n$ and $\tau$, all the paths from $\mathcal{D}\_{t\_n}$ to $\mathcal{\Theta}\_{n+1}$ and $\mathcal{B}\_{n+1}$ are blocked.** According to the graphical model properties, it shows the conditional independence; please see D-separation in graphical model literature [1]. Leveraging the conditional independence, we can derive the rule of computing $p(\mathcal{\Theta}\_{n+1} | \mathcal{D}\_{t\_n}\cup \mathcal{B}\_{n+1})$ --- it only needs the running posterior of the most recent states $\Theta\_{n}$ and $\tau$; we do not need to revisit $\mathcal{D}\_{t\_n}$. Please see Eq7 and below and Eq8 for the details. We can thereby keep tack the running posterior for each factor state, and integrate with the likelihood of the new data to compute the new running posterior (via CEP) --- that is how our online filtering algorithm works.

[1] Bishop, Christopher M., and Nasser M. Nasrabadi. *Pattern recognition and machine learning*. Chapter 8.3, Springer, 2006.

>C2: A simple baseline would tackle the problem as a regression task. It would be great to add it as a baseline.

R2: Great suggestion. We supplement the experimental results in the below. We used the entry indices and timestamps as features to build three regression models: Bayesian Linear Regression(BLR), RBF-kernel support vector machine (RBF-SVM), and a neural-network(NN) with 3 MLP layers. With the same training and test datasets, the predictive performance of these models is given in the below (we also list SFTL with R=5 for comparison).

| RMSE      | FitRecord | ServerRoom     |  BeijingAir-2 |     BeijingAir-3 |
| :---        |    :----:   |          ---: |---:  | ---:  |
| BLR      | 0.967 ± 0.003      | 1.164 ± 0.013   | 1.118 ± 0.016   | 1.074 ± 0.028  |
| RBF-SVM   |0.881 ± 0.012        | 1.232 ± 0.002      | 1.052 ± 0.018   | 1.018 ± 0.016   |
| NN   | 0.725 ± 0.024        | 1.431 ± 0.035      | 0.856 ± 0.054   | 0.871 ± 0.022   |
|SFTL-CP   | **0.424 ± 0.014**        | **0.161 ± 0.014**      | **0.248 ± 0.012**   | 0.473 ± 0.013   |
| SFTL-Tucker   | 0.430 ± 0.010       | 0.331 ± 0.056   |  0.303 ± 0.041   | **0.439 ± 0.019**  |

We can see SFTL outperforms all the regression models by a large margin, which confirms the advantage of SFTL and the necessity of learning factor representations. Note that SFTL also exhibits large improvement on other rank settings (see Table 3-5 of Appendix). We will supplement these results and a discussion into our paper.


>C3: Tensor Train or Tensor Ring decomposition not used

R3: Our current work only focuses on the classical yet perhaps the most widely used tensor decomposition forms, CP and Tucker, to prove our modeling and algorithmic framework are effective and promising. We definitely want to extend our method to more modern decomposition forms, like Tensor Train and Tensor Ring. However, it is non-trivial to adjust our current CEP inference to Tensor Train and Tensor Ring. The reason is that the chain of matrix multiplication in their decompositions makes the standard conditional moment matching intractable to compute. In the next step, we plan to develop novel computational tricks to address this issue so as to enable streaming factor trajectory learning for Tensor train and Tensor Ring decomposition as well.

---

### Author Response · Authors · 2023-08-14

In view of the limited available time, we would kindly like to ask the reviewers to please engage in a discussion with us (if not) given the submitted rebuttals so we can respond to the feedback in time.

---

### Comment · Area_Chair_Bvtp · 2023-08-17
**acknowledge and discuss the authors' detailed response**

The authors have provided a detailed response, which includes new experimental results more exposition of the technical material. *Drop a comment in this thread with how their response changed your perspective on the submission.*

Remember to also update your scores/reviews to reflect how your views have changed in light of the authors' response.

---

> ### Comment · Reviewer_Dugj · 2023-08-21
>
> Thank you for reminding us.
> I have read all the discussions and could not find any serious drawbacks to their results.
>
> > includes new experimental results
>
> I am satisfied with their original numerical results.
>
> > Tensor Train or Tensor Ring decomposition not used
>
> Because CP and Tucker models work well in their framework, I do not think we need to consider more modern models such as tensor-train and tensor ring. Their effectiveness with simple methods should be appreciated.

---

> > ### Comment · Reviewer_DKpV · 2023-08-22
> > **concluding comment**
> >
> > Thank you for the reminder.
> >
> > The authors' answers resolved the questions I had about the method.
> > The approach consists of several building blocks and parameters, and the authors validly argue for each choice.
> >
> > Their original numerical results on low-rank data make sense, and the authors promised to add experiments with more dense data.

---

### Decision · Program_Chairs · 2023-09-21

**Decision:**

Accept (poster)

**Comment:**

The paper introduces a family of dynamic tensor decomposition (CP or Tucker) models wherein the factors are time-indexed and and their trajectories are assumed to be drawn from a Gaussian (GP) process. Inference is challenging and the paper applies a clever trick to convert the GP prior into a state-space prior. The reviewers were impressed with the model and the convinced of the novelty and applicability of the techniques used to derive inference. The reviewers also engaged substantially with the authors during the discussion period and were all convinced by the authors' responses about their initial concerns and questions.